# Dimethylsulfide dynamics in first-year sea ice melt ponds in the Canadian Arctic Archipelago

Margaux Gourdal[1], Martine Lizotte[1], Guillaume Massé[1], Michel Gosselin[2], Michel Poulin[4], Michael Scarratt[3], Joannie Charette[2], Maurice Levasseur[1]

[1]Département de biologie, Québec-Océan and Unité Mixte Internationale 3376 TAKUVIK, CNRS-Université Laval, 1045 avenue de la Médecine, Québec, Québec G1V 0A6, Canada

[2]Institut des Sciences de la Mer de Rimouski (ISMER), Université du Québec à Rimouski, 310 allée des Ursulines, Rimouski, Québec G5L 3A1, Canada

[3]Maurice Lamontagne Institute, Fisheries and Oceans Canada, P.O. Box 1000, Mont-Joli, Québec G5H 3Z4, Canada

[4]Research and Collections, Canadian Museum of Nature, P.O. Box 3443, Station D, Ottawa, Ontario K1P 6P4, Canada

*Correspondence to*: Margaux Gourdal (margaux.gourdal@takuvik.ulaval.ca)

**Abstract.**

Melt pond formation is a seasonal pan-Arctic process. During the thawing season, melt ponds may cover up to 90% of the Arctic first-year sea ice (FYI), and 15 to 25% of the multi-year sea ice (MYI). These pools of water lying at the surface of the sea ice cover are habitats for microorganisms and represent a potential source of the biogenic gas dimethylsulfide (DMS) for the atmosphere. Here we report on the concentrations and dynamics of DMS in nine melt ponds sampled in July 2014 in the Canadian Arctic Archipelago. DMS concentrations were under the detection limit ($< 0.01$ nmol l$^{-1}$) in freshwater melt ponds, and increased linearly with salinity ($r_s = 0.84$, $p \leq 0.05$) from $\sim 3$ up to $\sim 6$ nmol l$^{-1}$ (avg. $3.7 \pm 1.6$ nmol l$^{-1}$) in brackish melt pond. This relationship suggests that the intrusion of seawater in melt ponds is a key physical mechanism responsible for the presence of DMS. Experiments were conducted with water from three melt ponds incubated for 24h with and without the addition of two stable isotope-labelled precursors of DMS (dimethylsulfoniopropionate) (D6-DMSP) and dimethylsulfoxide ($^{13}$C-DMSO). Results show that de novo biological production of DMS can take place within brackish melt ponds through bacterial DMSP uptake and cleavage. Our data suggest that FYI melt ponds could represent a reservoir of DMS available for potential flux to the atmosphere. The importance of this ice-related source of DMS for the Arctic atmosphere is expected to increase as a response to the thinning of sea ice and the areal and temporal expansion of melt ponds on Arctic FYI.

# 1 Introduction

Melt ponds represent an important but understudied component of the Arctic sea ice system. Snow deposited at the surface of the sea ice progressively melts during the thawing season and may accumulate above sea level in depressions at the surface of the ice to form melt ponds (Lüthje et al., 2006), likely through a recently identified process of percolation blockage (Polashenski et al., 2017). In the Arctic, melt pond fraction over first-year sea ice (FYI) in late spring-summer usually ranges from 50 to 60%, locally reaching 90% (Fetterer and Untersteiner, 1998; Eicken et al., 2004; Lüthje et al., 2006; Perovich et al., 2011). Rösel et al. (2012) have reported a 15% increase of the relative melt pond fraction for the month of June during the last decade (2001-2011) in the Arctic, most likely attributable to global climate change. This partly reflects the progressive replacement of multi-year sea ice (MYI) by FYI observed since the 1980's (National Snow and Ice Data Center, NSIDC, http://nsidc.org), favouring the formation of shallow melt ponds that spread over increasingly large areas (Agarwal et al., 2011; Ehn et al., 2011). The importance of melt ponds in the Arctic, as a water-air interface involved in heat and gas exchanges, is thus expected to increase in the future.

Dimethylsulfide (DMS) is the main natural source of reduced sulfur for the atmosphere (Bates et al., 1992). Between 17.6 to 34.4 Tg of sulfur are released annually from the ocean to the atmosphere (Lana et al., 2011), accounting for 50-60% of the natural reduced sulfur emitted (Stefels et al., 2007). DMS is also a climate-relevant gas potentially involved in a feedback loop known as the "CLAW" hypothesis (Charlson et al., 1987) linking biology and climate through the production of DMS-derived sulfate aerosols. According to CLAW, DMS emissions may affect the global radiation budget directly through the scattering of incoming solar radiation, and indirectly via the production of cloud condensation nuclei (CCN) leading to the genesis of longer-lived clouds with higher albedo (Twomey, 1974; Albrecht, 1989). Inspiring three decades of research and hundreds of publications, the feedback mechanism proposed by Charlson et al. (1984) remains yet to be demonstrated in its entirety (e.g. Ayers and Cainey, 2008). Although modelling results show that DMS emissions may have a negative radiative effect (e.g. Bopp et al., 2004; Gunson et al., 2006; Thomas et al., 2010), CCN may exhibit a low sensitivity to changes in DMS on a global scale (Woodhouse et al., 2010). Recent studies questioning the relative importance of DMS in new particle formation have emerged, suggesting that the global CLAW feedback may be weak (e.g. Quinn and Bates, 2011; Green and Hatton, 2014). On a regional scale however, the response of CCN production to change in DMS may vary by a factor of 20 (Woodhouse et al., 2010). The impact of DMS emissions on cloud properties (through the production of CCN) could be particularly important in remote pristine marine areas such as the polar regions (Carslaw et al., 2013). In the Southern Ocean, DMS may have contributed up to 33% of the increase in CCN observed south of 65°S as a response of increased wind speed since the early 1980s (Korhonen et al., 2010). The summertime Arctic marine boundary layer (MBL) is left relatively clean after seasonal wet deposition of particles and reduced atmospheric transport of aerosols from anthropogenic sources at lower latitudes (Stohl, 2006; Browse et al., 2012; Croft et al., 2016). Such pristine conditions, combined with thermally stable MBL are typical of the Arctic summertime (e.g. Aliabadi et al., 2016). Clean Arctic air masses allow ultrafine (5 - 20 nm diameter) particle formation (Burkart et al., 2016), and the potential growth of secondary

marine organic aerosols (including DMS-derived particles) into CCN (Willis et al., 2016). Hence, the Arctic is a favourable terrain for new particle formation from biogenic DMS (Chang et al., 2011; Rempillo et al., 2011; Collins et al., 2017; Giamarelou et al., 2016; Mungall et al., 2016; Willis et al., 2016).

DMS stems mainly from the enzymatic cleavage of dimethylsulfoniopropionate (DMSP) by algal and bacterial DMSP-lyases. DMSP is a cellular metabolite found in several phytoplankton species as particulate DMSP ($DMSP_p$) (see the review of Green and Hatton, 2014). $DMSP_p$ plays various roles in phytoplankton, including osmoregulation (Lyon et al., 2016), cryoprotection (Karsten et al., 1996), and prevention of cellular oxidation (Sunda et al., 2002). Part of the $DMSP_p$ produced by algae is released in the water column as dissolved DMSP ($DMSP_d$) via several pathways reviewed in Stefels et al. (2007), including active exudation, cell lysis, viral lysis and zooplankton grazing. $DMSP_d$ is then readily available to heterotrophic bacteria as carbon and sulfur sources (Kiene et al., 2000; Simó, 2001; Vila-Costa et al., 2006). The fraction of $DMSP_d$ consumed by heterotrophic bacteria and enzymatically cleaved by DMSP-lyases into DMS (DMS yield) may vary depending on the composition of microbial communities, their sulfur requirements, and the availability of other reduced forms of sulfur (Kiene et al., 2000; Stefels et al., 2007). DMSP-lyase are also present in several members of the microalgal groups Haptophyceae and Dinophyceae, and to a lesser extent Chrysophyceae (Niki et al., 2000). Ultimately, between ~1 and 40% of the DMSP produced by algae reaches the atmosphere as DMS (Stefels et al., 2007; Simó and Pedros-Alio, 1999a). In addition to the DMSP enzymatic cleavage pathway, DMS production may arise from dimethylsulfoxide (DMSO) reduction by various groups of marine bacteria including proteobacteria (e.g. Vogt et al., 1997), members of the Roseobacter group (González et al., 1999) and mat-forming cyanobacteria (van Bergeijk and Stall., 1996). However, the ubiquity of this DMSO-to-DMS reduction pathway amongst bacterial assemblages has not been established (Hatton et al., 2012). A limited number of phytoplankton species could also be involved in the reduction of DMSO into DMS (e.g. Fuse et al., 1995; Spiese et al., 2009). Increasing evidence suggests that particulate DMSO ($DMSO_p$) may be directly synthesized by a potentially wide range of marine phytoplankton and could be involved in osmoprotection, cryoprotection (Lee and de Mora 1999), and anti-oxidant protective mechanisms (Sunda et al., 2002). As for dissolved DMSO ($DMSO_d$), it is ubiquitous in seawater and continuous improvements in analytical techniques suggest that $DMSO_d$ may be as abundant as DMS in surface waters (e.g. Simó et al., 2000). DMSO is also a known sink for DMS (Hatton et al., 2004) via bacterial and photo oxidation of DMS to DMSO. Vertical Mixing and ventilation are also major removal processes influencing DMS concentrations in surface mixed layers (Bates et al., 1994; Kieber et al., 1996; Simó and Pedrós-Alió, 1999b; del Valle et al. 2007, 2009).

Ice-associated environments such as bottom sea ice, brine channels, melt ponds, under-ice surface waters, and leads provide complex and dynamic habitats to diverse microorganism communities involved in sulfur cycling (Levasseur, 2013). In the Arctic, the highest microalgal biomasses are found in the bottom ~0.1 m of sea ice, with Chlorophyll *a* (Chl *a*) concentrations several orders of magnitude above values for under-ice waters values (e.g. Legendre et al. 1992). A similar pattern of DMSP, DMSO and DMS build-up in bottom ice has been reported both in the Arctic and Antarctica (Kirst et al., 1991; Levasseur et al. 1994; Turner et al., 1995; DiTullio et al., 1995; Lee et al., 2001; Trevena et al., 2003; Trevena and

Jones 2006; Delille et al., 2007; Tison et al., 2010; Asher et al., 2011; Nomura et al., 2012; Galindo et al., 2015). For example, $DMSP_p$ concentrations up to 15 000 nmol $l^{-1}$ have been documented during spring in bottom FYI of the Eastern Arctic (Galindo et al., 2014) while surface water concentrations generally range between $1\pm0.2$ and $50\pm29$ (Vila-Costa et al., 2008; Matrai et al., 2007). DMSP, DMSO and DMS are also present throughout the ice column within the brine network (Levasseur et al., 1994; Trevena and Jones, 2006; Asher et al., 2011). Given that primary producers are the sole source of DMSP, very high ice concentrations of Chl $a$ can be correlated with DMSP through a first order relationship (Levasseur, 2013). This Chl $a$ to DMSP relationship may not hold for lower biomass concentrations (Tison et al., 2010). In addition to the variability induced by inter-specific differences in DMSP cellular contents (e.g. Keller et al., 1989; Stefels et al. 2004), environmental forcing are known to control DMSP, DMSO and DMS concentrations. In ice-associated environments, brine volume fraction might also be key in explaining DMS cycling variability via the control of ice permeability (Carnat et al., 2014). Structural changes within sea ice during the melt season, namely increases in brine volume fraction and ice desalination, result in increased connectivity and permeability in the warming sea ice (Willis et al., 2016; Polashenski et al., 2012) and influence DMSP and DMS cycling (Tison et al., 2010; Carnat et al., 2014). Also, phytoplankton blooms developing under the ice during the melting period have been shown to produce large quantities of $DMSP_p$, potentially leading to a build-up of DMS concentrations (Levasseur et al., 1994). In spite of the spatial importance of melt ponds, only few studies have investigated their role as a source of DMS for the Arctic atmosphere (e.g. Levasseur, 2013; Nomura et al., 2012).

Considerable efforts have been dedicated to the understanding of underlying process controlling the physics of melt ponds and their feedbacks on climate through the control of surface energy balance of the ice (Lüthje et al., 2006; Polashenski et al., 2017). However, little is known about their biogeochemistry. Four studies have specifically reported on DMS in melt ponds so far. They reveal negligible DMS concentrations in MYI ice melt ponds in the Central Arctic Ocean, and concentrations up to 2.2 nmol $l^{-1}$ in the High Arctic (Leck and Persson, 1996; Sharma et al., 1999). In Antarctica, DMS concentrations ranging between 1.1 and 3.7 nmol $l^{-1}$ and between below the detection limit (d.l.) and 250 nmol $l^{-1}$ were measured in two studies (Nomura et al., 2012 and Asher et al. 2011, respectively). In the latter study, bacterial DMSO reduction was suggested as a possible mechanism responsible for the high DMS concentrations observed although no actual rates of DMS production, either from DMSO or DMSP, were measured. High DMS concentrations reported in the Antarctic are most likely related to the development of a surface ice community following flooding, a process whereby heavy snow load pushes the ice below the water level. Flooding is common in the Antarctic and results in the formation of snow ice (Hunke et al., 2011). Several studies document melt pond colonization by micro-, nano- and pico-sized algae as well as bacteria (Bursa, 1963; Gradinger et al., 2005; Elliott et al., 2015), suggesting that DMS in melt ponds may originate from algal and bacterial metabolism. Yet, in situ DMS production had never been measured nor had key mechanisms been identified. Here, we report on the DMS concentrations in nine melt ponds located in the Eastern Canadian Arctic Archipelago (CAA), and on the prerequisites and processes responsible for the presence of this climate-active gas. This is the

first attempt to assess the dynamics of DMS in Arctic melt ponds. We identified sea ice permeability as a major control of DMS production in melt ponds, mediating the transport of both DMS and DMS-producing communities toward the surface of sea ice. We also provide the first evidence for direct in situ DMS production in Arctic melt ponds. We propose that seasonally melting sea ice might become increasingly prone to DMS production as FYI become largely predominant at the regional scale.

## 2 Materials and Methods

### 2.1 Study sites and environmental measurements

Nine melt ponds distributed between four stations located in Navy Board Inlet (Ice1 - MP1 and MP2 – 18 July), Barrow Strait (Ice2 - MP1 to MP3 – 20 July, and Ice3 - MP1 and MP2 – 21 July), and Resolute Passage (Ice4 - MP1 and MP2 – 23 July) were sampled during the joint NETCARE/ArcticNet research cruise conducted in 2014 on board the Canadian Coast Guard Ship (CCGS) *Amundsen* (Fig. 1).

At each station except for Ice2 (logistical constraints associated with ship time line), measurements of sea ice thickness, snow depth and sea ice freeboard (the height of sea ice above the ocean surface), were conducted within a 3 m distance of the melt ponds using a gauge (Kovacs Enterprise, Roseburg, OR, USA) (Table 1). The 3 m distance was a compromise between maximizing the proximity of ice and melt pond samples and minimizing melt pond disturbance during sampling operations. Ice and freeboard thickness presented in table 1 are averaged values of the seven (Ice1) to eight (Ice3 and Ice4) ice cores sampled at each station between the team members for their respective projects. In order to estimate the permeability of the ponded ice, sea ice temperature and bulk salinity were measured following Miller et al. (2015) at stations Ice1, Ice3 and Ice4. Two ice cores for sea ice temperature and salinity measurements were extracted using a 0.09 m core barrel (Kovacs Mark II, Kovacs Enterprise, Roseburg, OR, USA). In situ sea ice temperature profiles were measured directly, at 0.1 m intervals, using a high-precision thermometer (Testo® 720; precision of $\pm 0.1°C$). Corresponding sea ice salinity profiles were also determined at 0.1 m intervals. Each 0.1 m section was cut with a handsaw, stored in a plastic container, and allowed to melt at room temperature. Bulk salinity of the melted ice section was determined using a conductivity probe (Cond 330i, WTW™; precision of $\pm 0.1\%$). Permeability to fluid transport was assessed with brine volume profile calculations from bulk salinities and sea ice temperatures following equations from Leppäranta and Manninen (1988) for sea ice temperatures > -2ºC (Fig. 3). Due to logistical constraints mentioned above, neither ice nor snow measurements were conducted at station Ice2.

Melt pond depth, length and width were determined using a graduated stick and a tape ruler. Melt pond water temperature was measured using a high precision thermometer (61220-601 digital data logger, VWR) and water salinity was measured using the conductivity probe mentioned in the previous paragraph (Table 2). For each sampling location, two to

three members of the research team visually assessed the pond fraction based on pictures taken from the bridge (see Fig. 1c for examples) and a mean value was calculated.

## 2.2 Phytoplankton biomass and enumeration, bacterial count

For Chl $a$ quantification, 1000 ml to 1500 ml duplicates of in situ pond water were filtered onto Whatman® GF/F 25 mm filters. Pigments were extracted in 90% acetone for 18 to 24 h in the dark at 4°C (Parsons et al., 1984). Fluorescence of the extracted pigments was measured on board with a Turner Designs fluorometer (model 10-005R; Turner Designs, Inc.) before and after acidification with 5% HCl. The fluorometer was calibrated with a commercially available Chl $a$ standard (*Anacystis nidulans*, Sigma). Chl $a$ concentrations were calculated using the equation provided by Holm-Hansen et al. (1965).

Microscopic identification and enumeration of eukaryotic cells > 2μm were conducted in each melt pond. Samples of 250 ml were collected and preserved with acidic Lugol solution (0.4% final concentration; Parsons et al., 1984), then stored in the dark at 4°C until analysis was conducted by inverted microscopy (Lund et al., 1958, Parsons et al., 1984). For each sample, a minimum of 400 cells (accuracy ± 10%) and three transects of 20 mm were counted at a magnification of 400x. The main taxonomic references used to identify the eukaryotic cells are Tomas and Hasle (1997), Bérard-Therriault et al., (1999) and Throndsen et al. (2003).

The abundance of bacteria was determined by flow cytometry (Marie et al., 2005). Duplicate 4 ml subsamples were fixed with 20 μl of 25% glutaraldehyde Grade I (0.1% final concentration; Sigma-Aldrich G5882), then subjected to quick-freeze in liquid nitrogen for 24h, and finally stored at -80°C until analysis. Samples were analyzed using a FACS Calibur FCB3 flow cytometer (Becton Dickinson). Heterotrophic bacteria samples were stained with SYBR Green I and measured at 525 nm to quantify bacteria with Low Nucleic Acid (LNA; potentially less active) and High Nucleic Acid (HNA; potentially more active) content (Gasol and del Giorgio 2000, Lebaron et al. 2001). Analysis were performed on an Epics Altra flow cytometer (Beckman Coulter), fitted with a 488 nm laser (15 mW output; blue), using Expo32 v1.2b software (Beckman Coulter).

## 2.3 DMS(P) sampling, conservation and analysis

Duplicate samples for total DMSP (DMSP$_t$), dissolved DMSP (DMSP$_d$) and DMS measurements were collected from the melt ponds using a submersible pump (Cyclone – Aquameric™) connected to a sealed Lead-Acid battery and fitted with LDPE tubing. The pump was placed close to the pond bottom, without touching the ice. Stratification was reported in open melt ponds (i.e. melt ponds that have melted all the way to the sea surface) in Arctic FYI (Jung et al., 2015). However, closed FYI melt pond, such as those sampled during this study, are not prone to vertical stratification due to convective- and

wind-driven- mixing (Skyllingstad and Paulson, 2007). Given their shallow depths (less than 0.3 m), melt pond stratification was most probably inexistent or minimal during our study. Glass serum bottles were filled with sampled water, temporarily sealed with a butyl cap and an aluminum lid, and kept in the dark in a cooler until analysis upon return to the ship. Analysis were performed using a purge and trap (PnT) system coupled to a Varian™ 3800 gas chromatograph (GC), equipped with a

Pulsed Flame Photometric Detector (PFPD). Analytical precision of the method was better than 5%. Analytical detection limit (d.l.) was 0.01 nmol $l^{-1}$ for all sulfur compounds. The protocol is a modified version of the method of Leck and Bågander (1988) as described in Scarratt et al. (2000) and further revised in Lizotte et al. (2012). Briefly, DMS was stripped from liquid samples using helium gas (Praxair™ He, purity 99.999%) flowing at $50 \pm 5$ ml $min^{-1}$ in the PnT system. One to 5 ml of sample was injected in the PnT. Five ml of MilliQ™ water (Millipore filter system, Millipore Co., Bedford, MA,

USA) were subsequently pushed into the system to completely flush the sample into the glass bubbling chamber. The outer walls of the bubbling chamber were heated at 70°C with a circulating bath. Humidity in the gas sample downstream of the bubbling step was minimized using a 4°C circulating bath to trigger condensation. A Nafion® membrane separated the gas sample and He-carrier gas from a drying He counter-flow set at 100 ml $min^{-1}$ to further desiccate the gas sample. Fluxes in the PnT system were monitored using a flowmeter (Varian™).

For $DMSP_t$ samples, 3.5 ml of melt pond water was collected in duplicate into a 5 ml Falcon™ tube. $DMSP_d$ samples were obtained using the less disruptive Small-Volume gravity Drip Filtration (SVDF) method (Kiene and Slezak, 2006). Particulate DMSP ($DMSP_p$) concentrations were calculated by subtracting $DMSP_d$ from $DMSP_t$. DMSP samples were preserved with 50 µl of 50% sulfuric acid ($H_2SO_4$) to prevent DMSP transformation and remove pre-existing DMS. Samples were analyzed using the same methods as described above for DMS samples, following mole-to-mole conversion of DMSP

into DMS via NaOH (5 M) hydrolysis (Dacey and Blough, 1987).

## 2.4 Process studies

In order to examine the pathways of in situ DMS production in melt ponds, three 24h incubation experiments were conducted with water from the MP1 sampled at stations Ice1, Ice3, and Ice4. Water from the melt ponds was collected using

the pump described in sect. 2.3, pooled in clean 19 litres Coleman™ cooler jugs on site, and then transferred into gas-tight 3 litres polyvinyl fluoride Tedlar® bags. Light transmittance through the incubation bag material diminished with decreasing light wavelength. Between 99 to 92% of the photosynthetically active radiations (PAR, 400-700 nm) were transmitted through the bag material. Transmittances of Ultraviolet A radiations (UVA, 315-400 nm), and Ultraviolet B radiations (UVB, 290-315 nm) ranged between 92 to 82%, and 82 to 38%, respectively. The incubation bags were rinsed once with

~10% HCl, three times with MilliQ™ water, and twice with melt pond water to avoid contamination. The bags were custom-built and pre-closed on three sides (Dalian Delin Gas Packaging Co., Ltd.). After the addition of the melt pond water, the

bags were sealed with Clip-n-seal™ Teflon closure devices. A valve was fitted to each bag to allow the removal of any remaining bubbles.

The samples were subjected to three duplicate treatments (total of 6 bags): 1) two bags of unaltered melt pond water incubated under natural light (Control), 2) two bags amended with D6-DMSP and $^{13}$C-DMSO (100 nmol l$^{-1}$, final concentration each) incubated under natural light (Light-DMSP/O or L-DMSP/O), and 3) two bags amended with D6-DMSP and $^{13}$C-DMSO (100 nmol l$^{-1}$, final concentration each) incubated in the dark (Dark-DMSP/O or D-DMSP/O). L- and D-DMSP/O bags were amended with ~100 µl of freshly thawed aliquots of two D6-DMSP and $^{13}$C-DMSO stock solutions (high purity >99%, Sigma-Aldrich®). The high concentrations of isotopes added aimed to trigger a rapid and clear biological response (i.e. potential DMS production rates) measurable during our 24h incubations. DMSP and DMSO uptake are not expected to be mutually exclusive and have been observed concomitantly both in live cultures (Spiese et al., 2009) and in situ (Asher et al., 2011).

Bags were incubated on the foredeck of the ship. The temperature was kept as near to in situ water temperature as possible by continuously flowing surface seawater in the incubator. The temperatures of the incubation water for Ice1-MP1, Ice3-MP1 and Ice4-MP1 were $1.29 \pm 1.75°C$, $-0.28 \pm 0.26°C$, and $-0.73 \pm 0.09°C$, respectively. These mean values were within 1°C of the in situ melt pond water temperatures (Table 2).

DMSP$_t$, DMSP$_d$ and DMS concentrations were measured in duplicate every 6h during the incubation period as described above. DMS production from DMSP cleavage and DMSO reduction were determined through GC/mass spectrometry (MS) analysis as an increase of D6-DMS and $^{13}$C-DMS, respectively, in the L-DMSP/O and D-DMSP/O Treatments. Discrimination by the microorganisms toward lighter (natural) isotopes of DMSP and DMSO is expected to be minimal (< 10%) according to Asher et al. (unpublished data). The observed rates of change in the concentration of DMS stable isotopes are thus assumed to be representative of the potential for DMS cycling in these melt ponds.

This experimental setup allows the measurement of the following rates over 6h and 24h : 1) net changes of in situ DMSP$_d$ and DMSP$_p$ in natural light derived from the difference of DMSP$_d$ and DMSP$_p$ concentrations versus time in the Controls, respectively, 2) net in situ microbial DMS production in natural light derived from the regression slope of DMS versus time in the Controls, 3) net potential DMSP$_d$ changes in natural light and in the dark derived from the regression slope of DMSP$_d$ versus time in L-DMSP/O and D-DMSP/O, 4) net potential DMS production rate in natural light and in the dark derived from the regression slope of DMS versus time in L-DMSP/O and D-DMSP/O. The daily rates were obtained from the slopes between final and initial concentrations over 24h. Our experimental setup also allows the estimation of the relative contribution of DMSP and DMSO to the production of DMS, using the discrimination of the different isotopes of DMS (see sect. 2.5).

## 2.5 DMS isotopic signatures

The discrimination of the different isotopic forms of DMS, including D6-DMS and [13]C-DMS stemming from D6-DMSP cleavage and [13]C-DMSO reduction, respectively, was performed using GC-MS analysis following purging as described hereafter. Two sets of DMS sample duplicates were taken for the incubation experiments. The first set of duplicates was measured directly on-board using the Varian™ 3800 GC described in sect. 2.2. The second set of DMS duplicates was preserved through cryo-trapping. Cryo-trapping of DMS was conducted using glass GC liners filled with Tenax-TA polymer (high sulfur affinity) (Pio et al., 1996; Zemmelink et al., 2002; Pandey and Kim, 2009) kept at -80°C prior to their use, and maintained below -10°C during the 5 minute purging and trapping process. The Tenax-filled deactivated liners were mounted downstream of the PnT system described earlier. After gas extraction from the liquid samples, Tenax liners and their DMS content were wrapped individually in aluminum foil, placed in a Pyrex™ glass tube sealed with a Teflon lid, and returned to the -80°C freezer for several weeks until analysis on a land-based GC-MS.

Quantification of D6-DMS and [13]C-DMS was conducted via GC-MS analysis (6978 GC coupled to a 7000B Triple-Quad MS from Agilent). Mass spectra were collected both in full scan (m/z 45–100) and in selected ion monitoring (m/z 62, 63 and 68) modes. Final concentrations were calculated from standard curves using known concentrations of both unlabelled DMS and labelled DMS carrying the D6-DMS and [13]C-DMS signatures. The comparison between fresh DMS samples measured directly on-board during the NETCARE/ArcticNet campaign and cryo-preserved DMS samples shows excellent agreement between the two methods ($r^2 = 0.96$, Fig. 2).

## 2.6 Satellite data

Distances between each stations and the open ocean were assessed using scaled NASA's Earth Observing System Data and Information System (EOSDIS) imagery. Maps of the ice cover were accessed for the sampling dates in July 2014 through the MODIS (Terra/Aqua) Corrected Reflectance (True Color) layer combined with MODIS (Terra) Corrected Reflectance (Bands 3,6,7). These data are accessible in open source through the Global Imagery Browse Services (GIBS) (https://worldview.earthdata.nasa.gov). The imagery had a resolution of 250 m on a daily scale.

## 2.7 Statistical Analysis

Normality of the data was assessed using the Shapiro-Wilk test with a 0.05 significance level (R statistical software, R Core Team, 2016), which revealed that most variables were non-normally distributed (n=9, df=8, $\alpha=0.05$). Non-parametric Spearman's rank correlation test ($r_s$) with a 0.05 significance level was used to assess correlation between key variables since normality could not be achieved uniformly through standard normalization methods. Model I linear regressions ($r^2$) were used to determine biological rates during the incubation experiments (Sokal and Rohlf, 1995).

A non-parametric Mann-Whitney U test was used to determine whether the distributions of reduced-sulfur compounds (i.e. DMS, $DMSP_p$ and $DMSP_d$) in the Ice1-MP1 and Ice4-MP1 incubations experiments were statistically different from one another. The difference in reduced sulfur compound concentrations between the two incubation experiments was not found to be statistically significant (n=45, df=16 $\alpha$=0.05).

Based on the results of the Mann-Whitney U test, a series of Wilcoxon Signed-rank tests with a significance level $\alpha$=0.05 were conducted on the combined datasets of stations Ice1-MP1 and Ice4-MP1 in order to 1) assess the presence of statistical differences between the Controls and each Treatment L-DMSP/O and D-DMSP/O; 2) assess the potential effect of light on the concentrations and change rates of the reduced sulfur compounds under study (DMS, $DMSP_d$ and $DMSP_p$) by comparing paired dependent samples (repeated measures) from L-DMSP/O and D-DMSP/O.

# 3 Results

## 3.1 Ponded sea ice and snow properties

The physical characteristics of the sea ice surrounding the melt ponds are presented in table 1 and in figure 3. All the sampling sites were characterized by FYI, which was the predominant ice type throughout the region under study. Averaged sea ice thickness around the melt ponds were relatively uniform, varying between $1.13 \pm 0.07$ and $1.27 \pm 0.01$ m at the different sites. Average freeboard values were relatively more variable. Station Ice1 was characterized by low ice freeboards $-0.01 \pm 0.01$ m. Station Ice3 had the highest positive freeboards with $0.10 \pm 0.02$ m. Station Ice4 freeboards were also positive and showed the greatest variability, with $0.07 \pm 0.04$ m.

Brine volume fraction was calculated using sea ice salinity and temperature values, and used as a proxy of sea ice permeability (Fig. 3). Averaged values for bulk sea ice salinity over the full thickness of the ice were 1.73, 2.83 and 3.75 at stations Ice1, Ice3 and Ice4, respectively. Maximum bulk salinity never exceeded 5.00 (Ice4, 1.2-1.3 m section). In situ temperatures, averaged over the full thickness of the ice, were -0.54 °C, -0.52 °C and -0.98 °C at stations Ice1, Ice3 and Ice4, respectively, and reached a minimum value of -1.39 °C (Ice4, 0.8-0.9 m section). Brine volume fraction constantly exceeded 10% in the ice profiles, except in the upper 0.1 m section of the Ice3 station, where we likely observed the effects of refreezing metamorphosis of snow and/or sea ice recrystallization. Snow meltwater percolation and refreezing can form such superimposed ice layers as observed at station Ice3. The resulting impermeable layer at the top of the ice contributed to the high freeboard (0.1 m) measured at this station, representing 6% of the total ice thickness. Visual estimates of the pond fraction ranged from 30 to 60% (see Fig. 1c) and the remaining surface of sea ice was bare ice at stations Ice1, Ice2 and Ice4.

## 3.2 Physical, chemical and biological characteristics of the melt pond water

The physical and chemical characteristics of the melt ponds are presented in table 2. All melt ponds were closed melt ponds, i.e. not directly connected with the water column (Lee et al., 2012). The mean depth of the individual melt ponds ranged from 0.07 to 0.29 m, with length and width varying between 1.00 and 25.00 m (Fig. 1). Melt pond water temperatures and salinities varied between 0.21 and 1.86°C and between 0.2 and 8.5, respectively. Chl $a$ concentrations were variable, ranging from 0.03 to 0.48 µg l$^{-1}$ with a mean of 0.20 µg l$^{-1}$ (Table 3). The composition of the algal assemblage present in the melt ponds will be described in detail in a companion paper (Charette et al., *in prep.*) but is summarized in table 3. The algal assemblages were dominated by unidentified flagellates, ice-associated pennate diatoms, and chrysophytes. Empty diatom frustules were abundant in all melt ponds. Abundance of heterotrophic bacteria with high nucleic acid content (HNA) varied between 0.02 and 0.24 × 10$^9$ cells l$^{-1}$ (Table 3).

In situ DMSP$_p$ and DMSP$_d$ concentrations ranged from 1.8 to 4.0 nmol l$^{-1}$, and from below d.l. (< 0.01 nmol l$^{-1}$) to 1.4 nmol l$^{-1}$, respectively. Melt pond DMS concentrations ranged from below d.l. to 6.1 nmol l$^{-1}$ (Table 3). Spearman's rank correlation coefficients between key in situ variables measured in the melt ponds are presented in table 4. DMS concentrations significantly co-varied with salinity ($r_s = 0.84$, $p < 0.05$) and Chl $a$ ($r_s = 0.84$, $p < 0.05$). None of the other variables measured displayed significant relationships between each other (not shown).

**3.3 Dynamics/cycling of reduced sulfur compounds in Arctic melt ponds**

Results from the Ice1-MP1 and Ice4-MP1 incubation experiments are presented in Fig. 4 (4a-c left and 4b-d right, respectively). Results from the Ice3-MP1 experiments are not presented since DMSP$_d$ and DMS concentrations showed no variation during the 24h incubation period in the Controls and in the Amended Treatments. This will be discussed in sect. 4.2.2.

During the Ice1-MP1 incubation, initial DMSP$_d$ concentration was 1.30 nmol l$^{-1}$ in the Control and slightly increased to reach 5.3 nmol l$^{-1}$ during the 24h incubation period (Fig. 4a). In the Light (L-DMSP/O) and Dark (D-DMSP/O) Amended Treatments, DMSP$_d$ concentrations started at 102 nmol l$^{-1}$, decreased to $\sim 35$ nmol l$^{-1}$ at T$_6$, and remained stable (Dark Treatments) or decreased to 10 nmol l$^{-1}$ (Light Treatments) until T$_{24}$ (Fig. 4a). Concentrations of DMS in the Control of Ice1-MP1 started at 3.0 nmol l$^{-1}$, increased to 8.8 nmol l$^{-1}$ between T$_0$ and T$_6$, and then decreased regularly to 4.2 nmol l$^{-1}$ at T$_{24}$ (Fig. 4c). The addition of labelled DMSP and DMSO stimulated DMS production. In the L-DMSP/O Treatment, DMS concentrations increased to 12.6 nmol l$^{-1}$ at T$_6$, remained at this level between T$_6$ and T$_{12}$, increased again between T$_{12}$ and T$_{18}$ and remained stable at $\sim 19$ nmol l$^{-1}$ between T$_{18}$ and T$_{24}$ (Fig. 4c). DMS concentrations were consistently higher in the D-DMSP/O Treatment than in L-DMSP/O (Fig. 4c). They first reached 15.6 nmol l$^{-1}$ at T$_6$, increased gradually to reach a peak value of 24.2 nmol l$^{-1}$ at T$_{18}$, and decreased slightly to 21.6 nmol l$^{-1}$ at T$_{24}$. Note that dissolved DMSO was not measured during this study due to methodological issues.

In the Ice4-MP1 incubation, $DMSP_d$ concentrations started at 3.0 nmol $l^{-1}$ in the Control and remained close to this value during the whole experiment (Fig. 4b). In the L-DMSP/O and D-DMSP/O Amended Treatments, $DMSP_d$ concentrations started at 87 and 96 nmol $l^{-1}$, respectively. As observed in the previous melt pond, the concentrations decreased to ~45 nmol $l^{-1}$ at $T_6$, and then slowly decreased to a value of ~30 nmol $l^{-1}$ at $T_{24}$ (Fig. 4b). DMS concentrations in the Control of Ice1-MP1 started at 2.6 nmol $l^{-1}$ and remained at this level during the 24h experiment (Fig. 4d). In the L-DMSP/O Treatment, DMS concentrations increased more or less linearly from 2.6 nmol $l^{-1}$ at $T_0$ to 6.7 nmol $l^{-1}$ at $T_{24}$. In the D-DMSP/O Treatment, the increase in DMS concentrations was more pronounced than in the Light Treatment, and a maximal value of 11.5 nmol $l^{-1}$ was reached at $T_{24}$.

In situ and potential change rates of the sulfur compounds during the incubation experiments are presented in tables 5 and 6, respectively. Changes in $DMSP_d$, and to a lesser extent DMS concentrations were generally not linear over the 24h incubation period, with more pronounced variations during the first 6 h. To take into account this non-linearity, both hourly rates measured between $T_0$ - $T_6$ and $T_6$ - $T_{24}$, as well as daily rates ($T_0$ - $T_{24}$) are presented in these tables.

In Ice1-MP1, the concentrations of $DMSP_p$ in the Control decreased at a rate of 2.2 nmol $l^{-1} d^{-1}$ (Table 5). We measured no change in $DMSP_d$ during the first 6 h, but a positive net increase of 4.0 nmol $l^{-1}$ over the full 24h incubation period was observed. In situ DMS changes increased by 1.0 nmol $l^{-1} h^{-1}$ during the first 6 h and by 1.2 nmol $l^{-1} d^{-1}$ over 24 h. Potential net $DMSP_d$ change rates of -11.6 and -10.2 nmol $l^{-1} h^{-1}$ were measured during the first 6 h of incubation in L- and D-DMSP/O Treatments, respectively (Table 6). These rates became -1.2 and -0.6 nmol $l^{-1} h^{-1}$ between $T_6$ and $T_{24}$ in L- and D-DMSP/O, respectively. Over 24 h, negative potential net $DMSP_d$ change rates of ~ -91 nmol $l^{-1}$ and -71 nmol $l^{-1}$ for the L-DMSP/O and D-DMSP/O Treatments were calculated. Positive potential net DMS change rates of 1.6 and 2.1 nmol $l^{-1} h^{-1}$ were measured during the first 6 h of incubation in L-DMSP/O and D-DMSP/O, respectively. For the complete 24h incubation, potential net DMS change rates reached 15.4 nmol $l^{-1} d^{-1}$ in the Light and 18.6 nmol $l^{-1} d^{-1}$ in the Dark.

In Ice4-MP1, in situ $DMSP_p$ decreased at a rate of 1.9 nmol $l^{-1} d^{-1}$ over the course of the incubation (Table 5). Meanwhile, in situ $DMSP_d$ changes rates were below the d.l. during the first 6 h and almost null over 24 h (Table 5). In situ DMS change rates were close to zero after 6 h, and below d.l. after 24 h. Potential net $DMSP_d$ change rates of -8.1 nmol $l^{-1} h^{-1}$ were measured during the first 6 h of incubation in both L- and D-DMSP/O (Table 6). These rates slowed down to -0.5 and -0.9 nmol $l^{-1} h^{-1}$ between $T_6$ and $T_{24}$, respectively. Over one day, average potential net $DMSP_d$ change rates of ~ -59 nmol $l^{-1}$ and -62 nmol $l^{-1}$ were calculated for the L-DMSP/O and D-DMSP/O Treatments. Potential net DMS change rates remained low in both L-DMSP/O and D-DMSP/O Treatments during the first 6 h of incubation with values at 0.1 and 0.3 nmol $l^{-1} h^{-1}$, respectively. For the complete 24h incubation, potential net DMS change rates in Light and Dark reached 4.2 and 8.9 nmol $l^{-1} d^{-1}$, respectively.

During both Ice1-MP1 and Ice4-MP1 incubation experiments, the Light versus Dark Treatment had no effect on the net changes in $DMSP_d$ concentrations between the L-DMSP/O and D-DMSP/O Treatments (Wilcoxon Signed-rank test; n=8,

df=3, α=0.05), but significantly impacted the rates of net accumulation of DMS (Wilcoxon Signed-rank test; n=12, df=5, α=0.05). The accumulation of DMS over 24h in the L-DMSP/O Treatments were consistently and significantly lower than in the corresponding D-DMSP/O Treatments (Wilcoxon Signed-rank test; n=8, df=3 ,α=0.05). Based on the difference between the L- and D-DMSP/O Treatments after 24 h, we estimated the light-associated DMS sinks at 3.2 nmol $l^{-1}$ $d^{-1}$ in Ice1-MP1 and at 4.7 nmol $l^{-1}$ $d^{-1}$ in Ice4-MP1 (Table 6).

### 3.4 Isotopic discrimination of DMS sources

Table 7 shows the concentrations of DMS isotopes (m/z 62) and (m/z 68) after 24h incubation in the three treatments and their relative contribution (%) to the total DMS measured at $T_{24}$. As expected, 100% of the total DMS in the Controls of the two experiments (3.0 nmol $l^{-1}$ and 2.3 nmol $l^{-1}$) showed the isotopic signature of natural DMS (m/z 62). In the L-DMSP/O Treatment of the Ice1-MP1 incubation, 78% (14.4 nmol $l^{-1}$) of the DMS measured at $T_{24}$ derived from D6-DMSP additions (m/z 68), with the remaining 22% (4.1 nmol $l^{-1}$) being natural DMS (Table 5). Similarly, 73% (18.2 nmol $l^{-1}$) of the DMS measured at $T_{24}$ derived from D6-DMSP additions in the D-DMSP/O Treatment, with the remaining 27% (6.6 nmol $l^{-1}$) carrying the signature of natural DMS.

In Ice4-MP1, 80% (5.1 nmol $l^{-1}$) of the DMS measured at $T_{24}$ in the L-DMSP/O Treatment derived from the added D6-DMSP, with the remaining 20% (1.3 nmol $l^{-1}$) carrying the signature of natural DMS. For the D-DMSP/O Treatment, 65% (7.9 nmol $l^{-1}$) of the DMS at $T_{24}$ derived from the D6-DMSP addition with 35% (4.2 nmol $l^{-1}$) originating from natural DMS. The absence of (m/z 63) DMS, regardless of the treatment, indicates that $^{13}$C-DMSO reduction was not contributing to the production of DMS during these two experiments (m/z 63 not shown in table 7). The match between the sum of DMS isotopes (m/z 62 and m/z 68) and the total fresh DMS concentration measured on board (Fig. 2) also confirms the absence of DMSO-to-DMS reduction during our experiments.

### 4 Discussion

Research on DMS dynamics in melt ponds is in its infancy. Before this study, only four publications reported DMS measurements in melt ponds, two in the Arctic (Leck and Persson, 1996; Sharma et al., 1999) and the two others in the Antarctic (Asher et al., 2011; Nomura et al., 2012). In the Arctic, Leck and Persson reported negligible levels of DMS in MYI melt ponds while Sharma et al. (1999) measured concentrations reaching 2.2 nmol $l^{-1}$. In the Antarctic, Nomura et al. reported DMS concentrations inferior to 3.7 nmol $l^{-1}$ while Asher et al. (2011) measured levels up to 250 nmol $l^{-1}$. Our results show that DMS concentrations in Arctic melt ponds may be at least three times higher (up to ~6 nmol $l^{-1}$) than the first Arctic measurements and that both physical and biological processes can contribute to the accumulation of this climate-active gas

in these transient environments. As discussed hereafter, evidences suggest that different ice cover dynamics and microbial communities are the two probable leading causes for the reported variability in DMS concentrations between melt ponds.

## 4.1 Physical controls of DMS concentrations in melt ponds

The strong relationship observed between DMS concentrations and salinity in the melt ponds sampled ($r_s = 0.84$, $p \leq 0.05$, Table 4) suggests that salinization processes may play a crucial role in the initial seeding of DMS (and probably DMS-producing microbial assemblages) and the resulting cycling of DMS within melt ponds. Three main mechanisms could be involved in the salinization of closed melt ponds: 1) deposition of sea spray from the ice margin/leads, 2) brine intrusion, and 3) seawater intrusion through porous/low freeboard sea ice. For the reasons explained below, seawater intrusion through

porous/low freeboard sea ice appears to be the most likely mechanism responsible for the salinization of the melt ponds during our study.

Sea spray probably did not contribute significantly to the salinization of the melt ponds during our study. The salinization of melt ponds could occur through sea spray deposition or seawater overflow during stormy events. Sea spray can transport salts over distances ranging from a few meters for the largest particles to a maximum distance of $\sim 30$ km for

finer aerosols, depending on wind speed (McArdle and Liss, 1995). This requires favourable wind direction, a relative proximity of the melt ponds with open water areas, and as demonstrated hereafter regarding the melt ponds studied here, unrealistic volumes of sea spray. During our study, the average volume of the melt ponds was 8 m³. We conservatively estimated that 19 to 367 litres of sea spay (assuming an average sea surface salinity of 33) was required to increase melt pond salinity from zero to 0.2 or 8.5, as measured during our study. Considering both the relatively large volume of sea

spray required and the far-reaching distances (>15 km, estimated from MODIS data) of the sampled melt ponds from open water at the time of sampling, sea spray was unlikely the main source of salt in the melt ponds studied.

Ice brine intrusion is also unlikely to have contributed significantly to melt pond salinization since the averaged bulk ice salinity was low (under 5), and locally did not exceeded 2 (top 0.2 m). It is also known that most of the hyper-saline brine characterizing consolidated cold FYI in winter are lost in spring through full depth brine convection well before melt

ponds start to form (Jardon et al., 2013). Residual salts are finally lost through meltwater flushing during the summer season (Weeks and Ackley, 1986, Eicken et al., 2002; Vancoppenolle et al., 2007). At the time of our sampling, low bulk salinity values, combined with calculated brine volume fraction constantly exceeding 10% in the entire sea ice profiles (except in the upper 0.1 m section of the Ice3 station) suggest that full depth flushing had already occurred. We thus exclude sea ice brine enrichment of melt ponds as their main salinization mechanism.

Rather, we suggest that melt ponds salinization originated mostly from the intrusion of seawater through the ice. Although closed melt ponds are not visibly connected to seawater, exchanges with the underlying seawater can take place. The extent of these exchanges are dependent on the sea ice freeboard and micro-structure, i.e. the amount, size and shape of

brine inclusions (Carnat et al., 2014), that controls sea ice permeability. Above a critical brine volume ranging between 5% (for columnar sea ice) and 10% (for granular sea ice), brine inclusions become interconnected. During the melting season, decrease in sea ice thickness is enhanced by the formation of the melt pond and lead to a loss of freeboard. As melt ponds become closely levelled with seawater, small changes in ice temperature oscillating around the freezing temperature may result in episodic intrusion of seawater mixed with meltwater through the porous ice. Seawater mixed with meltwater entering the brines channels of permeable sea ice may bring salts, nutrients and microorganisms (Jardon et al., 2013, Vancoppenolle et al., 2010), potentially reaching surface melt ponds. This mechanism most probably explains the salinity and biochemical characteristics of Ice1 and Ice4 melt ponds. Station Ice3 represents a different case. Here, the low melt pond salinity (and absence of biological activity) may be explained by the presence of an impermeable ice layer on the top of the ice preventing both pond drainage and exchange between pond water and seawater.

We acknowledge that our data set is too limited to draw firm conclusions on the processes leading to the formation and salinization of FYI melt ponds. Yet, in the interest of further research, we conjecture that snow load before melt onset may be crucial in determining the fate of melt ponds not only with regards to their saline status, but also their potential to produce DMS. Brine volume, derived from bulk salinity and temperature, generally provides a valid proxy for sea ice permeability. In some case however, melting of high snowpack generates a considerable flow (up to 15cm $d^{-1}$) of freshwater into the porous structure of sea ice (Polashenski et al., 2017). This can create localized ice plugs within the highly connected brine network of apparently porous sea ice and allow melt ponds to persist above sea level well after sea ice bulk sea ice brine volume reached a critical level (5-10%). Such deviation from the porosity/permeability relationship following freshwater intrusion is demonstrated in Polashenski et al. (2017). We suggest that we observed such case of melt pond persistence above sea level in station Ice3. Alternatively, lower snow load remaining at the onset of the melt season will translate into a less abundant freshwater input above sea ice. Snow load distribution is however notoriously highly variable even at the meter scale due to wind redistribution and sea ice topography variability (e.g. Polashenski et al., 2017). Low snowpack would induce limited insulation of the sea ice from atmospheric conditions, resulting in 1) a more gradual warming of sea ice during spring season, and 2) limited freshwater loading available for percolation blockage. In this case, freshwater would not seal the ice through percolation blockage (Polashenski et al., 2017). Sea ice would then remain entirely porous as soon as the 5-10% brine volume threshold is reached, facilitating melt pond salinization process. We suggest that this scenario may have been observed at stations Ice1 and Ice4.

## 4.2 Biological control of DMS production in melt ponds

### 4.2.1 Simulated in situ conditions

In addition to the physical mechanisms mentioned above, results from our incubation experiments show that biological production of DMS may take place in Arctic melt ponds under simulated in situ conditions, and to a higher extent following DMSP enrichment. A daily net DMS production of 1.2 nmol $l^{-1} d^{-1}$ was measured without substrate addition in one of the

three melt ponds tested, Ice1-MP1 (Table 5). The absence of net daily increase in DMS in the two other melt ponds tested does not necessarily preclude potential gross production since, as discussed below, this production could be balanced by microbial DMS uptake and photolysis. Such balance between DMS sources and sinks over a 24h period has been previously observed during incubation experiments conducted with Labrador Sea water (Wolfe et al., 1999). However, this explanation probably does not explain the absence of accumulation of DMS in the freshwater melt pond Ice3-MP1 since the addition of substrate failed to stimulate DMS production (see sect. 4.2.3).

### 4.2.2 Source of DMS under substrate amended conditions

Bacterial $DMSP_d$ metabolism was the main mechanism leading to DMS production in the melt ponds tested. None of the DMS measured carried the (m/z 63) isotopic signature that would have indicated its $^{13}C$-DMSO origin. Extremely high gross DMS production rates from DMSO reduction, up to $105 \pm 24$ nmol $l^{-1}$ $d^{-1}$, were measured within Antarctic sea ice brines by Asher et al. (2011). The authors suggested that this mechanism could also potentially be responsible for the high DMS concentrations (up to 250 nmol $l^{-1}$) measured in Antarctic melt ponds. The absence of DMS production from $^{13}C$-DMSO in the melt ponds studied here may then reflect potential differences in microbial assemblages within melt ponds, as the metabolic ability to convert DMSO into DMS is not ubiquitous among bacterial communities (Hatton et al., 2012). In support of this hypothesis, it has been shown that between 70 and 78% of the operational taxonomic units (OTU), a marker of microbial diversity, in Arctic and Southern Ocean surface water communities are unique to their region (Ghiglione et al., 2012). Observed differences in the biological characteristics of melt ponds between the poles could also reflect divergent sea ice dynamics. Antarctic sea ice salinity is higher by 0.5 to 1.0% than in Arctic sea ice (Gow et al., 1982, 1987) and the C-shaped salinity profile that is typical in fully formed Arctic FYI is not as prominent in Southern Ocean sea ice (Eicken, 1992). Antarctic sea ice is commonly subjected to intense rafting, flooding, and the formation of snow ice (Hunke et al., 2011). Antarctic melt ponds studied in Asher et al. (2011) may have been subjected to flooding leading to the formation of salted "freeboard layers" (Haas et al., 2001; Massom et al., 2006). This hypothesis is highly plausible considering that highest salinities were reported in the top sea ice layers and salinity decreased throughout the ice profile. Such configuration may bring productive microbial communities at the surface of the ice, potentially responsible for the high DMS concentrations observed in melt ponds. The still limited availability of data, including other published studies, prevents us from firmly conclude further on the specific reasons of the absence of DMS production from $^{13}C$-DMSO and compels additional exploration.

### 4.2.3 Substrate limitation of microbial DMSP uptake and DMS production

The addition of DMSP had a strong stimulating effect on the bacterial uptake of DMSP and the resulting production of DMS in the two brackish melt ponds tested. In both Ice1-MP1 and Ice4-MP1, the response of the microbial assemblage to the

addition of DMSP was rapid and intense (Fig. 4) as approximately half of the $DMSP_d$ added was consumed over the first 6 h and potential net DMS production increased substantially.

In the Amended Treatments, changes in the $DMSP_d$ concentrations over time proceeded into two distinct phases during the incubation period (Figs. 4a-c). Irrespective of the light regime, the first phase ($T_0$ to $T_6$) was characterized by a rapid net decrease of $DMSP_d$ concentrations. Potential net $DMSP_d$ change rates of $\sim$ -11 nmol $l^{-1}$ $h^{-1}$ and -8.1 nmol $l^{-1}$ $h^{-1}$ in Ice1-MP1 and Ice4-MP1, respectively, were calculated (Table 6). These estimates represent minimum rates since our calculation assumes a linear uptake during the first 6 h. Even so, these rates already translate an extremely steep decrease of $DMSP_d$ in comparison with those of -0.01 to -0.2 nmol $l^{-1}$ $h^{-1}$ previously measured in the same region in the water column and under the ice cover in spring (Luce et al., 2011; Galindo et al., 2015). This difference most probably reflects the large amount of DMSP added in our experiments. The second phase of the incubation (from $T_6$ to $T_{24}$) shows an abrupt slowing down of the potential net $DMSP_d$ change rates, still slightly superior but closer to the range of in situ rates reported by the previous studies (Table 6). These results clearly show that an active microbial assemblage predisposed to $DMSP_d$ consumption inhabited the brackish melt ponds under study. This is in accordance with Sørensen et al. (2017) reported substrate limitation of bacterial growth in Arctic FYI melt ponds.

The bi-phasic DMSP uptake dynamics observed in our experiment suggests that DMSP additions at least temporarily fulfilled the microbial requirement for this substrate. Phytoplankton biomass, and probably dissolved organic carbon, was low in the melt ponds. In a context of substrate limitation, rapid uptake of $DMSP_d$ was expected. Fast and transient intracellular accumulation of compatible solutes, such as DMSP, may serve as an adaptive strategy by microbial cells to help cope with fluctuations of the surrounding environment, increasing their tolerance to osmotic and thermal stresses for example (Welsh, 2000). Such accumulations which could occur under replete conditions allow a so-called "luxury uptake" of compounds by microorganisms above their immediate requirements. Finally, the low HNA bacterial abundances measured in the melt ponds (Table 3) might explain the curtailing of $DMSP_d$ uptake measured after the initial rapid consumption.

Following $DMSP_d$ addition, the potential daily net DMS production rates varied between 4.2 and 18.6 nmol $l^{-1}$ in the two brackish melt ponds tested (Table 6). As previously mentioned, it was only within the freshwater Ice3-MP1 melt pond that potential to process DMSP and produce DMS was not detected, even when substrate limitation was alleviated by $DMSP_d$ addition. These different in situ and potential DMSP metabolisms and DMS production rates suggest that de novo DMS production in melt ponds is triggered only once a threshold in microbial biomass is reached. In support of this hypothesis, Chl $a$ concentration (0.05 µg $l^{-1}$) and bacterial abundance (0.02 x $10^9$ cells $l^{-1}$) were extremely low in the unproductive freshwater Ice3-MP1: one order of magnitude lower than in the two productive brackish Ice1-MP1 and Ice4-MP1 (Table 3).

In contrast with the simulated in situ conditions in the Controls, net potential DMS production in the Amended Treatments constantly exceeded DMS loss through photolysis and bacterial consumption, resulting in a net accumulation of

DMS throughout the 24 h of incubation (Fig. 4c, d). In spite of the atypically high DMSP level added, our $DMSP_d$ amendments could be considered as analogues of the $DMSP_d$ pulses that take place in the natural environment during the senescence phase of algal blooms, or under high viral attack and grazing pressure. These pulses are known to contribute to transient DMS build-up at lower latitudes (e.g. Malin et al., 1993; Locarnini et al., 1998; Scarratt et al., 2000). At high latitudes, the inhibitory effect of low temperature on microbial DMS consumption may even exacerbate these build-ups. For instance, temperatures below 2°C were found to potentially inhibit DMS consumption rates in the Labrador Sea (Wolfe et al., 1999). The sensitivity of DMS microbial uptake to low temperatures was proposed by Wolfe et al. (1999) as a potential driving mechanism responsible for the large pulses of DMS often measured in the Arctic environment. Cold and biologically active melt ponds may be prone to such DMS accumulation when the limitation in substrate is alleviated. However, our observations suggest that such events, that would require high biomass, may be rare in Arctic melt ponds.

### 4.2.4 Influence of light on DMSP bacterial metabolism

Light affected the accumulation of DMS in the DMSP/O Amended Treatments. The continuous light conditions prevailing during our incubation experiments reduced DMS accumulation in the L-DMSP/O Treatments compared to the D-DMSP/O Treatments by ∼15% and up to 40% in Ice1-MP1 and Ice4-MP1, respectively (Fig. 4c, d). This negative effect of light was expected since photolysis is known as an important sink for DMS in the open ocean, sometimes as important as bacterial consumption in the near surface waters (Royer et al., 2016). However, removing light did not increase $DMSP_d$ removal rates (Fig. 4a, b). It should be pointed out that our incubation setup did not aim to reproduce the exact light field of the melt ponds where light backscattering could considerably increase DMS loss by photolysis. The importance of light as a sink for DMS in melt ponds should be thoroughly investigated in future studies. Light-induced DMS losses may be particularly relevant in melt ponds since DMS ventilation, another important sink for DMS (absent from our incubation setup), is probably limited at least in small melt ponds where fetch is minimal.

### 5 Conclusion

Results from this study confirm the presence of DMS in Arctic melt ponds, with concentrations up to three times higher than those reported by the two other previous Arctic studies. Salinization of melt ponds appears to be a prerequisite to the presence of DMS and its de novo biological production. Intrusion of seawater through porous sea ice and low freeboard flooding seems to be a fundamental mechanism for bringing salt and DMS in the melt ponds as well as allowing the establishment of potential DMS-producing communities. As melt ponds become closely levelled with seawater, small changes in ice temperature oscillating around the freezing temperature may result in episodic intrusion of seawater mixed with meltwater through the porous ice. Seawater mixed with meltwater penetrating the brines channels of permeable sea ice may bring salts, nutrients and microorganisms potentially seeding surface melt ponds. Results from incubation experiments

reveal a modest but measurable in situ net production of DMS in one of the melt ponds tested. Evidence also suggests that melt ponds can host an active bacterial assemblage associated with rapid DMSP uptake when available and significant daily production of DMS. Freshwater ponds lacked the potential to produce DMS, further confirming the importance of the seawater intrusion mechanism in the biological cycling of DMS in melt ponds. No DMSO-to-DMS reduction was detected in our study.

To this day, most climatologies assume the absence of DMS fluxes above ice-covered waters (e.g. Lana et al., 2011) even though several studies provide direct (Zemmelink et al., 2008; Nomura et al., 2012, MYI) and indirect (Carnat et al., 2014, FYI) evidence of DMS venting from snow-covered Antarctic sea ice. Arctic studies have also reported DMS exchanges above the ice-covered ocean, specifically highlighting the importance of particular zones such as open leads (Levasseur et al., 1994) and cracks in sea ice, as well as melt ponds (Sharma et al., 1999; Mungall et al., 2016). Here, we measured an average DMS concentration of 2.1 nmol $l^{-1}$ (<0.01 nmol $l^{-1}$ – 6.1 nmol $l^{-1}$) in nine FYI melt pond. Although estimation of the actual DMS flux from the melt ponds sampled here is beyond the scope of our study, we argue that FYI melt ponds represent a non-negligible reservoir of DMS in the Arctic readily available for air-sea exchange. The estimation of the importance of melt ponds as net sources of DMS for the atmosphere will require an accurate evaluation of their spatial and temporal coverage, a better understanding of gas exchange between small fetch melt ponds and the atmosphere and its sensitivity to changing wind velocity, as well as comprehensive measurements of DMS within melt ponds at large, both FYI and MYI, and particularly at higher latitudes. How the strength of DMS emissions from melt ponds will respond to changes in Arctic climate is still unknown. Both the spatial extent of melt ponds and their temporal span have increased over the last three decades in connection with regional climate alterations (Stroeve et al., 2014; Agarwal et al., 2011). Meanwhile, MYI is increasingly being replaced by thinner FYI (e.g. Kwok et al., 2009), potentially promoting melt pond salinization processes through permeable sea ice. The importance of this ice-related source of DMS for the Arctic atmosphere could increase as a response of the structural changes of the Arctic ecosystem.

**Data availability**

Metadata are available on the Polar Data Catalog website at www.polardata.ca. Data are available on request by contacting the first author.

**Authors contribution**

Margaux Gourdal was responsible for the elaboration of the experimental design, the sampling process, the data analysis and processing, and the redaction of this paper. Several co-authors provided specific data included in the paper and all co-authors contributed to the final edition of the paper.

**Competing interests**

The authors declare that they have no conflict of interest.

**Special issue statement**

We are requesting your permission to link this paper to a NETARE BG special issue.

**Acknowledgements**

The authors wish to thank the commandant, officers, and crew of the Canadian ice-breaker NGCC *Amundsen* for their support during the project. The authors are especially indebted to Jean-Sébastien Côté, Tim Papakyriakou and Roghayeh Ghahremaninezhad for participating to the sample collection, Marjolaine Blais for pigment and bacterial abundance analysis, and Virginie Galindo for her extensive logistical support during the cruise. We thank Sylvie Lessard for cell identification. The authors would like to thank the reviewers for their valuable comments and suggestions to improve the quality of the paper. This project was funded by the Network on Climate and Aerosols: Addressing Key Uncertainties in Remote Canadian Environments (NETCARE), ArcticNet (Network of Centres of Excellence of Canada), the Canada Excellence Research Chair in Remote Sensing of Canada's New Arctic Frontier, and the Takuvik Joint International Laboratory. Partial funding was also provided by the Natural Sciences and Engineering Research Council of Canada (NSERC) and the Fonds de Recherche du Québec Nature et Technologies (FRQNT) through Québec-Océan. Funding support was also received from the Canadian Museum of Nature for cell counts analysis. The author received graduate scholarships from Université Laval, takuvik UMI and stipends from NETCARE and Québec-Océan.

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

**Figure captions**

Figure 1: (a) Regional map showing the location of the four sampling stations (Ice1 to Ice4) (red circles) during the NETCARE/ArcticNet 2014 campaign. (b) MODIS imagery above the four sampling station (red circles) showing the ice conditions on 18 July 2014 in the sampling area. (c) Left to right, pictures of stations Ice1, Ice2, Ice3 and Ice4 with size scale. MPF stands for the Melt Ponds Fraction visually estimated from the bridge for stations Ice1, Ice2, Ice3 and Ice4.

Figure 2: Relationship between the concentrations of fresh DMS samples measured on board the ship via gas chromatography during the campaign and the concentrations of the corresponding preserved duplicate samples measured via coupled gas chromatography and mass spectrometry in a laboratory setting. The concentrations of the preserved DMS samples plotted are the sum of the three isotopes of DMS investigated in this study (m/z of 62, 63, and 68; see Materials and Methods).

Figure 3: In situ temperature (●) and bulk ice salinity (○) profiles of the sea ice surrounding the melt ponds sampled at stations Ice1 (a), Ice3 (b) and Ice4 (c). Temperature and salinity values of each 0.1 m sea ice section were used to calculate brive volumes (▮), an indicator of sea ice permeability, throughout the full depth of sea ice (Cox and Weeks 1983, Petrich and Eicken 2010).

Figure 4: Temporal variations in $DMSP_d$ (a, c), and DMS (b, d) concentrations during the Ice1-MP1 and Ice4-MP1 incubation experiments. Both Light (○) and Dark (●) Treatments were initially amended with 100 nmol $l^{-1}$ of both D6-DMSP and $^{13}$C-DMSO. Control Treatments (△) mimic natural concentration changes over time. In (a) and (c), vertical bars represent standard errors of mean values between duplicate samples.

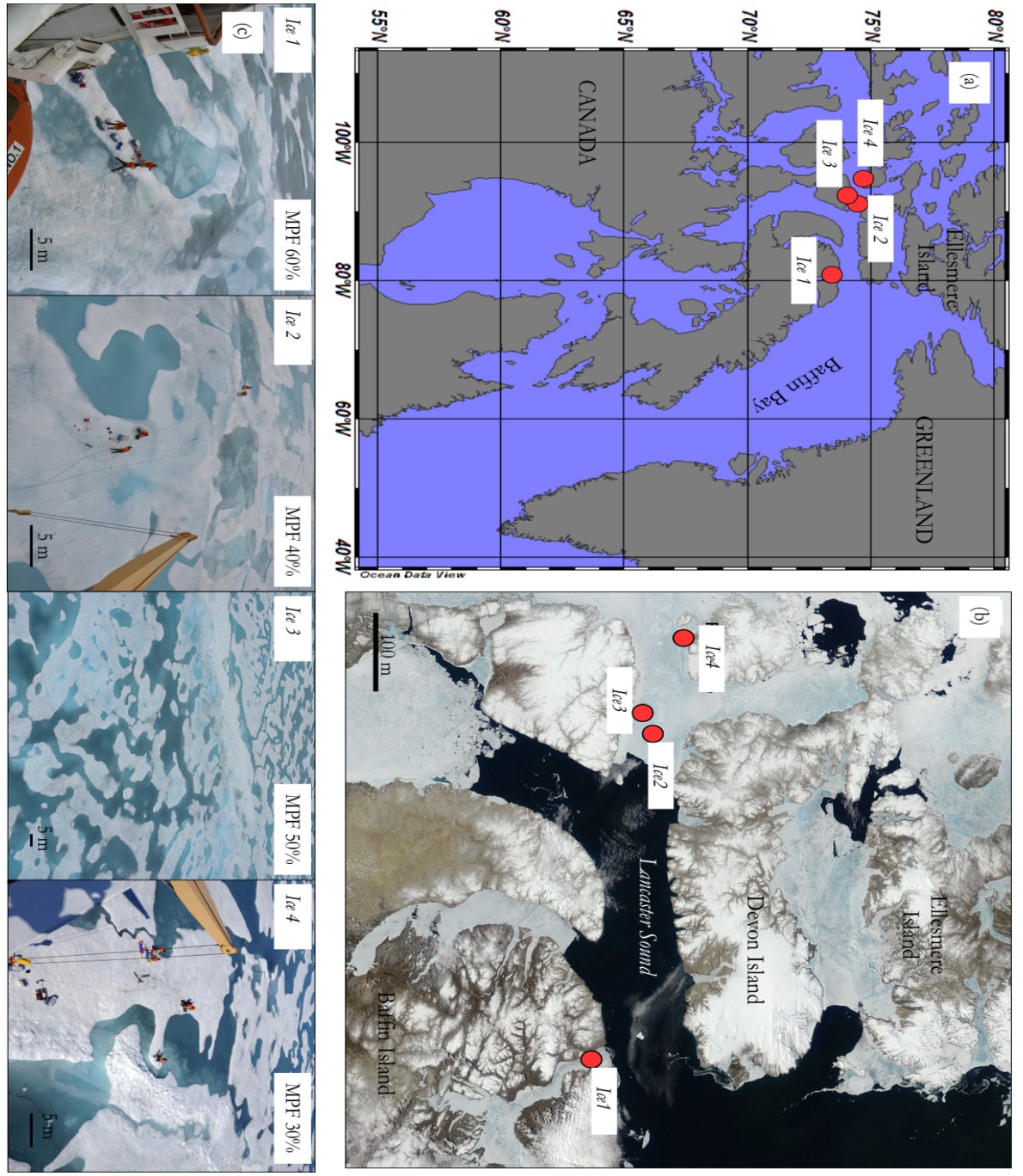

**Figure 1: (a)** Regional map showing the location of the four sampling stations (Ice1 to Ice4) (red circles) during the NETCARE/ArcticNet 2014 campaign. **(b)** MODIS imagery above the four sampling station (red circles) showing the ice conditions on 18 July 2014 in the sampling area. **(c)** Left to right, pictures of stations Ice1, Ice2, Ice3 and Ice4 with approximative size scale. MPF stands for the Melt Ponds Fraction visually estimated from the bridge for stations Ice1, Ice2, Ice3 and Ice4.

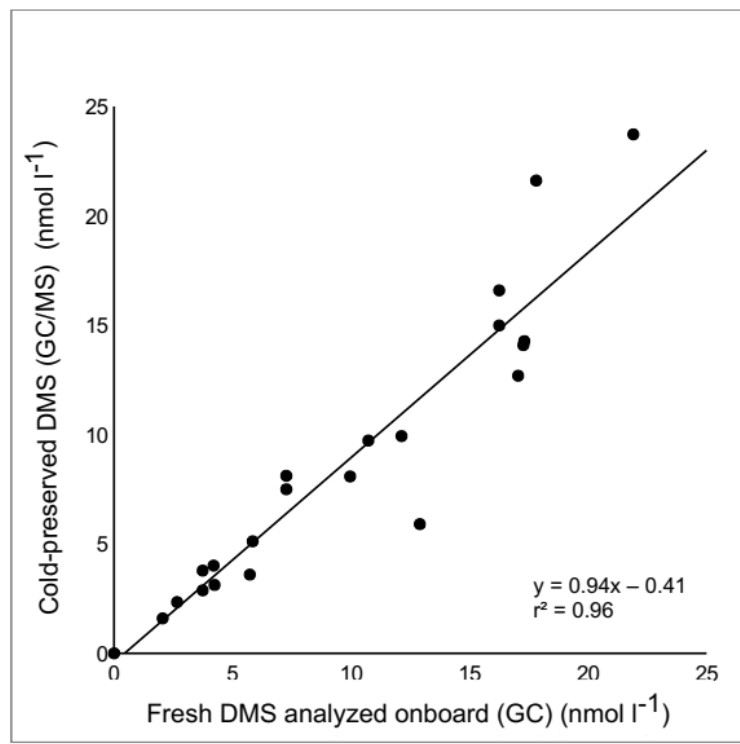

**Figure 2: Relationship between the concentrations of fresh DMS samples measured on board the ship via gas chromatography during the campaign and the concentrations of the corresponding preserved duplicate samples measured via coupled gas chromatography and mass spectrometry in a laboratory setting. The concentrations of the preserved DMS samples plotted are the sum of the three isotopes of DMS investigated in this study (m/z of 62, 63, and 68; see Materials and Methods).**

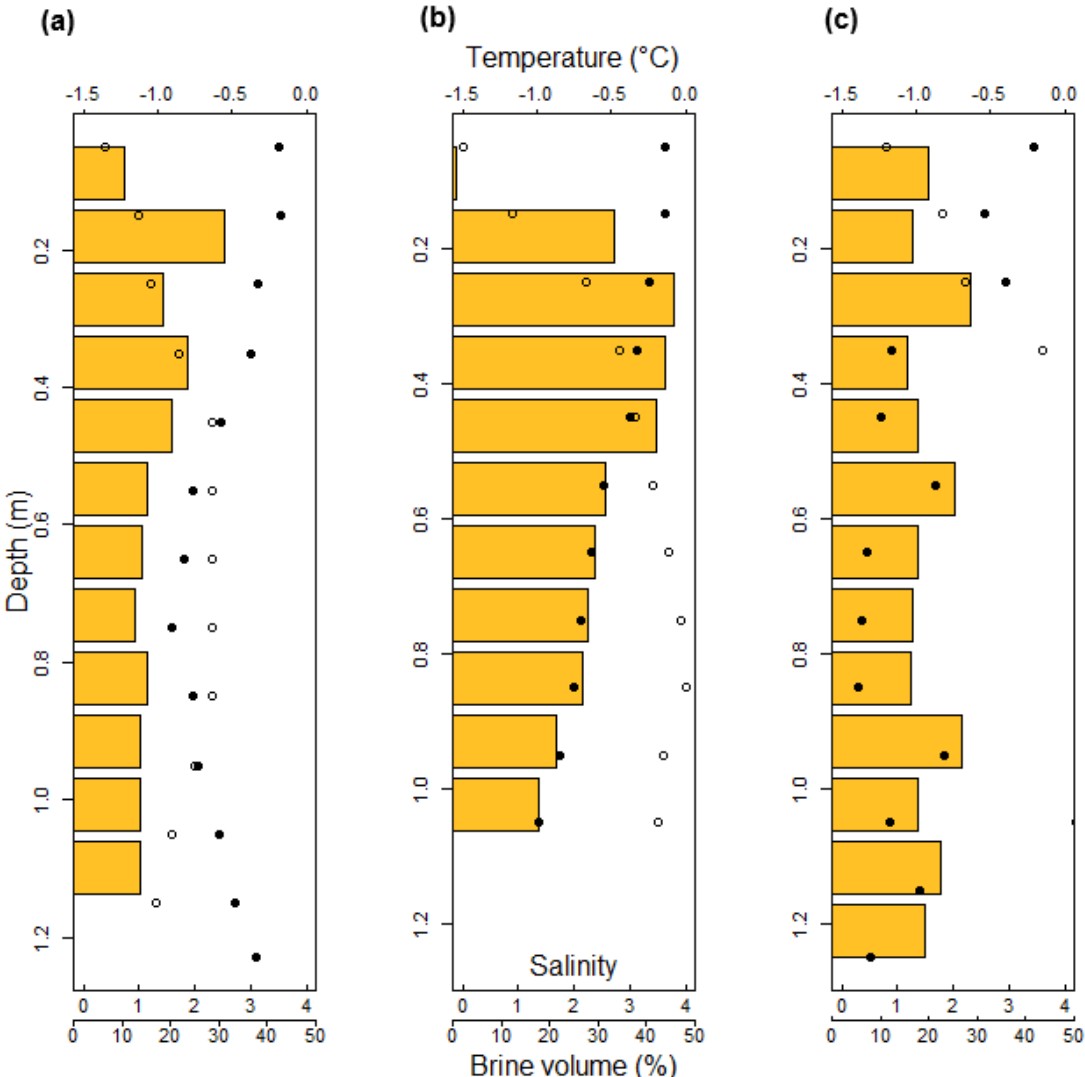

**Figure 3: In situ temperature (●) and bulk ice salinity (○) profiles of the sea ice surrounding the melt ponds sampled at stations Ice1 (a), Ice3 (b) and Ice4 (c). Temperature and salinity values of each 0.1 m sea ice section were used to calculate brine volumes (orange bars), an indicator of sea ice permeability, throughout the full depth of sea ice (Cox and Weeks 1983, Petrich and Eicken 2010).**

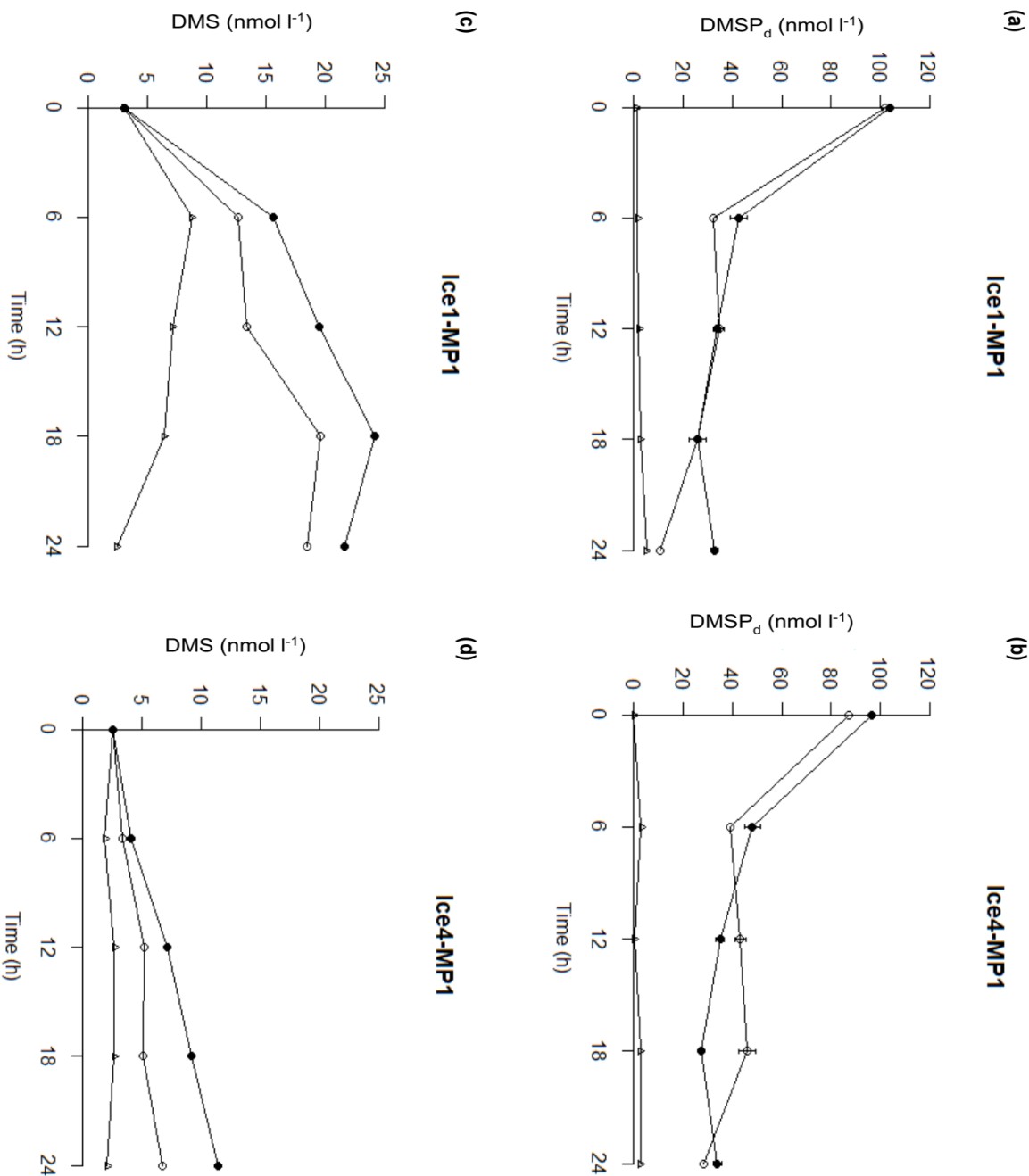

**Figure 4: Temporal variations in DMSP$_d$ (a, b), and DMS (c, d) concentrations during the Ice1-MP1 and Ice4-MP1 incubation experiments. Both Light (○) and Dark (●) Treatments were initially amended with 100 nmol l$^{-1}$ of both D6-DMSP and $^{13}$C-DMSO. Control Treatments (△) mimic natural concentration changes over time. In (a) and (b), vertical bars represent standard errors of mean values between duplicate samples. (Numbering order was modified as requested).**

**Table 1: Physical characteristics of the sea ice surrounding the melt ponds. Note that only melt pond sampling (i.e. no ice sampling) was conducted at station Ice2 due to ship-related logistical constraints. A negative freeboard height indicates that the ice surface was locally below the mean sea level. n/a stands for non-available data. Ice thickness and freeboard values are averages of 7 (Ice1) to 8 (Ice3 and Ice4) ice cores sampled at each station.**

| Station | Sampling date | Snow and frozen snow* depth (cm) | Ice thickness (cm) | Freeboard (cm) |
|---------|---------------|----------------------------------|--------------------|----------------|
| Ice1 | Jul 18, 2014 | 0 | 121 ± 2 | -1 ± 1 |
| Ice2 | Jul 20, 2014 | 0 | n/a | n/a |
| Ice3 | Jul 21, 2014 | 0 + 7* | 113 ± 7 | 10 ± 2 |
| Ice4 | Jul 23, 2014 | 0 | 127 ± 1 | 7 ± 4 |

**Table 2: Physical characteristics of the melt pond water. For melt pond depth, mean ± standard deviation values are presented.**

| Station | Melt pond # | Melt pond depth (m) | Melt pond salinity (psu) | Melt pond temperature (°C) |
|---------|-------------|---------------------|--------------------------|----------------------------|
| Ice1 | MP1 | 0.18 ± 0.01 | 5.2 | 1.9 |
| Ice1 | MP2 | 0.18 ± 0.04 | 4.1 | 1.8 |
| Ice2 | MP1 | 0.29 ± 0.05 | 0.7 | 0.4 |
| Ice2 | MP2 | 0.19 ± 0.03 | 0.4 | 0.3 |
| Ice2 | MP3 | 0.12 ± 0.01 | 0.2 | 0.2 |
| Ice3 | MP1 | 0.07 ± 0.01 | 1.1 | 0.2 |
| Ice3 | MP2 | 0.10 ± 0.00 | 0.9 | 0.2 |
| Ice4 | MP1 | 0.12 ± 0.01 | 8.1 | 0.2 |
| Ice4 | MP2 | 0.11 ± 0.02 | 8.5 | 0.3 |

**Table 3: Reduced sulfur compound concentrations measured in situ in the melt ponds and the associated biological characteristics (abundance of high nucleic acid (HNA) bacteria, Chl *a* concentrations, and relative abundances of major taxonomic groups) of the melt pond water.**

| Station | Melt pond | *In situ* DMSPp (nmol l⁻¹) | *In situ* DMSPd (nmol l⁻¹) | *In situ* DMS (nmol l⁻¹) | Abundance of bacteria (HNA) ( x 10⁹ cells l⁻¹ ) | Chl *a* (µg l⁻¹) | Abundance of algae ( x 10⁶ cells l⁻¹ ) | Dominant algal group |
|---|---|---|---|---|---|---|---|---|
| Ice1 | MP1 | 2.2 | 1.3 | 3.0 | 0.24 | 0.48 | 2.00 | Unidentified flagellates (50 %) Prasinophytes (ca. 25%) |
| | MP2 | 2.0 | 1.4 | 3.1 | | 0.40 | | Unidentified flagellates (55 %) Prasinophytes (ca. 25 %) |
| Ice2 | MP1 | 1.8 | d.l. | d.l. | 0.04 | 0.03 | 0.50 | Unidentified flagellates (90 %) Pennate diatoms (ca. 28 %) |
| | MP2 | 2.4 | d.l. | d.l. | | 0.09 | | Unidentified flagellates (50 %) Pennate diatoms (ca. 28 %) |
| | MP3 | 2.3 | d.l. | d.l. | | 0.06 | | Unidentified flagellates (70 %) Pennate diatoms (ca. 28 %) |
| Ice3 | MP1 | 2.0 | d.l. | d.l. | 0.02 | 0.05 | 0.30 | Unidentified flagellates (45 %) Chrysophytes (29 %) |
| | MP2 | 2.3 | d.l. | d.l. | | 0.04 | | Unidentified flagellates (55 %) Chrysophytes (23 %) |
| Ice4 | MP1 | 4.0 | d.l. | 2.6 | 0.15 | 0.18 | 1.00 | Unidentified flagellates (50 %) Pennate diatoms (20 %) |
| | MP2 | 3.7 | 1.1 | 6.1 | | 0.20 | | Unidentified flagellates (60%) Pennate diatoms (25 %) |

**Table 4: Spearman's rank correlation coefficients between key in situ variables measured in the melt ponds. * indicates a 0.05 significance level.**

|  | DMS | Salinity | Temperature | Chl *a* |
|---|---|---|---|---|
| DMS |  | 0.84* | 0.51 | 0.84* |
| Salinity |  |  | 0.40 | 0.56 |
| Temperature |  |  |  | 0.60 |

**Table 5: In situ DMSP$_p$, DMSP$_d$ and DMS change rates measured during the incubation experiments conducted in melt ponds Ice1-MP1 and Ice4-MP1. Hourly rates for DMSP$_d$ and DMS net changes measured between T$_0$ and T$_6$ as well as T$_6$ and T$_{24}$ are derived from the slope of DMSP$_d$ and DMS concentrations vs. time, respectively. Daily DMSP$_d$ change rates are calculated as the difference between the DMSP$_d$ concentrations measured at T$_{24}$ and T$_0$. Daily DMS change rates are calculated as the difference between the DMS concentrations measured at T$_{24}$ and T$_0$. Rates measured over the first 6 h and between T$_6$ and T$_{24}$ are expressed in nmol l$^{-1}$ h$^{-1}$. Other rates are expressed in nmol l$^{-1}$ d$^{-1}$.**

| | In situ DMSP$_p$ change rates | In situ DMSP$_d$ change rates | | In situ DMS change rates | |
| --- | --- | --- | --- | --- | --- |
| Station | (nmol l$^{-1}$ d$^{-1}$) | (nmol l$^{-1}$ h$^{-1}$) (6h) | (nmol l$^{-1}$ d$^{-1}$) | (nmol l$^{-1}$ h$^{-1}$) (6h) | (nmol l$^{-1}$ d$^{-1}$) |
| Ice1-MP1 | -2.2 | 0.0 | 4.0 | 1.0 | 1.2 |
| Ice 4-MP1 | -1.9 | 0.0 | -0.1 | -0.1 | 0.0 |

**Table 6: Potential net DMSP$_d$ change rates, potential net DMS change rates and light-associated DMS sinks measured during the incubation experiments conducted in melt ponds Ice1-MP1 and Ice4-MP1. Clear and shaded horizontal lines regroup the rates measured under natural light (L-DMSP/O) and in the dark (D-DMSP/O), respectively. Hourly rates for potential DMSP$_d$ and DMS net changes between $T_0$ and $T_6$ as well as $T_6$ and $T_{24}$ are derived from the slope of DMSP$_d$ and DMS concentrations vs. time, respectively. Daily potential net DMSP$_d$ change rates are calculated as the difference between the DMSP$_d$ concentrations measured at $T_{24}$ and $T_0$. Daily potential net DMS change rates are calculated as the difference between the DMS concentrations measured at $T_{24}$ and $T_0$. Rates of light-associated DMS sink were measured as the difference of DMS accumulation between L-DMSP/O and D-DMSP/O after the 24h incubation. Rates measured over the first 6 h and between $T_6$ and $T_{24}$ are expressed in nmol l$^{-1}$ h$^{-1}$. Other rates are expressed in nmol l$^{-1}$ d$^{-1}$.**

| Station | Potential net DMSP$_d$ change rates | | | Potential net DMS change rates | | Light-associated DMS sinks |
|---|---|---|---|---|---|---|
| | (nmol l$^{-1}$ h$^{-1}$) ($T_0$-$T_6$) | (nmol l$^{-1}$ h$^{-1}$) ($T_6$-$T_{24}$) | (nmol l$^{-1}$ d$^{-1}$) | (nmol l$^{-1}$ h$^{-1}$) ($T_0$-$T_6$) | (nmol l$^{-1}$ d$^{-1}$) | (nmol l$^{-1}$ d$^{-1}$) |
| Ice1-MP1 | -11.6 | -1.2 | -91.5 | 1.6 | 15.4 | 3.2 |
| | -10.2 | -0.6 | -71.3 | 2.1 | 18.6 | --- |
| Ice 4-MP1 | -8.1 | -0.5 | -59.2 | 0.1 | 4.2 | 4.7 |
| | -8.1 | -0.9 | -62.6 | 0.3 | 8.9 | --- |

**Table 7: (m/z 62) and (m/z 68) DMS concentrations after 24h incubation in the Control, L-DMSP/O and D-DMSP/O Treatments. Relative contribution (%) of natural DMS and D6-DMSP to the total DMS measured at $T_{24}$ in the Control, L-DMSP/O and D-DMSP/O Treatments during the incubation experiments with water from Ice1-MP1 and Ice4-MP1. Natural DMS signature = (m/z 62); signature of DMS derived from D6-DMSP = (m/z 68). No (m/z 63), which represents the signature of DMS derived from $^{13}$C-DMSO, was retrieved either after 12 h (not shown) or 24 h.**

| Incubation | Treatment | (m/z 62) DMS (nmol l$^{-1}$) | (m/z 68) DMS (nmol l$^{-1}$) | (m/z 62) % of total DMS | (m/z 68) % of total DMS |
|---|---|---|---|---|---|
| | Control | 3.0 | 0.0 | 100 | 0 |
| Ice1-MP1 | L-DMSP/O | 4.1 | 14.4 | 22 | 78 |
| | D-DMSP/O | 6.6 | 18.2 | 27 | 73 |
| | Control | 2.3 | 0.0 | 100 | 0 |
| Ice4-MP1 | L-DMSP/O | 1.3 | 5.1 | 20 | 80 |
| | D-DMSP/O | 4.2 | 7.9 | 35 | 65 |