# Peer review of "Dimethylsulfide dynamics in first-year sea ice melt ponds in the Canadian Arctic Archipelago"

_Biogeosciences, 2017_

## Referee Comment (RC1) · Anonymous Referee #1 · 4 Jan 2018

Review for "Dimethylsulfide dynamics in first-year sea ice melt ponds in the Canadian Arctic Archipelago" by Gourdal and co-authors

General comments: This is a generally well written and interesting manuscript describing novel measurements of Dimethylsulfide (DMS, DMSPd and DMSPp) concentrations and dynamics (derived from labelled DMSP and DMSO isotopic marker incubations) in Arctic sea-ice melt ponds. A shortcoming of the paper is that it is based on a rather limited dataset with consequent problems for statistical analyses. Only two (brackish) melt ponds very sampled for incubations in this study, and statistics are based on an N=2 (with additional duplicate - but apparently dependent - samples taken

from each incubation). While a t-test can be employed for a dataset with an N=2(4), the dataset appears extremely small to make any statistical relevant conclusions. This reviewer therefore suggests to clarify (provide df or define N values) or alternatively delete these statistical analyses and rephrase some of the statements in relation to DMSP transformation into DMS. This said, other methods applied in this study appear to be solid (noting that this reviewer is not an expert in GC/GC-MS DMS(P) analyses) and raise some important new research questions for future research on DMS dynamics in sea-ice melt ponds. In summary this reviewer suggests publication of the manuscript after amending the statistical analyses (t-test) and some other (minor) shortcomings including a re-consideration of the estimate of the overall DMS reservoir in Arctic melt-ponds, and a more detailed discussion on the sea-ice surface permeability.

Specific comments:

P2, L1: delete "natural" in first sentence of abstract, this word is not needed

P2 , L 12: This calculation of the DMS reservoir in Arctic melt-ponds is based on 2 single measurements of 2 very specific (=brackish) melt ponds in a very defined study area (e.g. the Canadian Archipelago). This reviewer considers up-scaling the results from this study to the entire Arctic as highly problematic. It is suggested to delete this estimate from the manuscript (see also page 16) or at least to delete this broad-brush estimate from the Abstract.

P4, L 17: No need to start a new paragraph

P4, L32: be more specific: . . .of melted ice samples" rather than "melt water samples"

P5, L1: The T and S data from the 10 cm surface ice allow the accurate calculation of the brine volume fraction according to established formulas, see e.g., Eicken, H., H. R. Krouse, D. Kadko, and D. K. Perovich, Tracer studies of pathways and rates of meltwater transport through Arctic summer sea ice, J. Geophys. Res., 107(C10), 8046,

doi:10.1029/2000JC000583, 2002; an references therein. Applying these formulas (e.g. those for high T and low S sea ice values, e.g. Manninen and Leppaeranta 1988, cited in above reference), and using the values reported in the manuscript of T = -0.2C and S = 0 psu actually indicates "im"-permeable ice, while a T = -0.2C and S = 0.8 psu indicates a brine volume of about 20% (= highly permeable ice). This reviewer suggest that brine volumes are calculated for the T – S measurements and that a more detailed discussion on ice permeability/sea water percolation is given. Please also note that a) "the rule of 5s" is primarily based on a brine volume fraction of 5% (which can be achieved by different T-S combinations, including T = -5C and S = 5, b) that this percolation threshold is only valid during thermodynamic equilibrium, and c) also only applies for columnar ice (likely the case in these samples), but this surface ice might have also undergone some melting/metamorphosis). In summary this reviewers suggest a more detailed discussion of the sea ice permeability. The current conclusions are fine, but just stating "according to the rule of 5s" is insufficient.

P4, L 11: ....replicates... How many?

P4, L25: This reviewer suggest to add a sentence and a definition of "HNA" here, e.g. what nucleic acid stain was used in this fly cytometry protocol?

P6, L23: It is unusual to refer to PAR as "700-400", normally one would write "400-700". This also applies to the UVA and UVB wavelengths given in the text.

P 8, L25: As discussed above, this reviewer suggests to revisit the t-test statistics applied: It appears that N equals 2, which makes application of the t-test problematic. At least more explanation is needed.

P 9, L3: This reviewer suggest to use the SI unit "m" rather than "cm" as unit for length measurements throughout the manuscript/figures.

P9, L7: As per above more details is required than just stating the "rule of fives".

P9, L 16: use singular, e.g. "detail"

P11, L10 -15: If "significantly" is used test-statistics should be given, also provide df value and/or N. Given the low N, these statistical results are of little relevance.

P 12, L 17: Sea "spray" rather than "spay"

P12, L28: Here "gravity drainage" and "brine flushing" are used to describe the same process, while classically "brine drainage" refers to the release of cold salt brines in surface-cooled sea ice, while "brine flushing" refers to the flushing out of salt through meltwater, e.g. they are technical terms used for different physical processes.

P 13, L 10: No data are shown that demonstrate :"full depth desalinization" -> please clarify

P 13, L 20: Avoid the use of "significant" if no statistical test was conducted /or provide statistical results.

Fig and Tables:

Fig 3: Unusual numbering of panels: "c" should be "b" and "b" should be "c"?

Tab 7: "control" or "Control" -> consistency in spelling needed

---

## Author Comment (AC1) · 24 Jan 2018

Referee #1: This is a generally well written and interesting manuscript describing novel measurements of Dimethylsulfide (DMS, DMSPd and DMSPp) concentrations and dynamics (derived from labelled DMSP and DMSO isotopic marker incubations) in Arctic sea ice melt ponds. A shortcoming of the paper is that it is based on a rather limited dataset with consequent problems for statistical analyses. Only two (brackish) melt ponds were sampled for incubations in this study, and statistics are based on an N=2 (with additional duplicate - but apparently dependent - samples taken from each incubation). While a t-test can be employed for a dataset with an N=2(4), the dataset

appears extremely small to make any statistical relevant conclusions. This reviewer therefore suggests to clarify (provide df or define N values) or alternatively delete these statistical analyses and rephrase some of the statements in relation to DMSP transformation into DMS. This said, other methods applied in this study appear to be solid (noting that this reviewer is not an expert in GC/GC-MS DMS(P) analyses) and raise some important new research questions for future research on DMS dynamics in sea ice melt ponds. In summary this reviewer suggests publication of the manuscript after amending the statistical analyses (t-test) and some other (minor) shortcomings including a re-consideration of the estimate of the overall DMS reservoir in Arctic melt-ponds, and a more detailed discussion on the sea ice surface permeability.

Author's response to general comments: We thank the reviewer for his/her positive general evaluation of the paper and helpful comments. The following actions were taken:

-We acknowledge that our restricted dataset limited the power of the chosen statistics analyses. We thus removed the results of the Student's t-test (P8, L25) and used a non-parametric Mann-Whitney U test (a replacement for independent groups t-test) that allowed us to compare the two independent groups of samples from the Ice1-MPI and Ice4-MP1 incubation experiments. The Mann-Whitney U test revealed no significant differences between the distributions of the reduced-sulfur compounds (i.e. DMS, DMSPd and DMSPp) from the Ice1-MP1 and Ice4-MP1 incubation results (n=45, df=16, $\alpha$=0.05). The conclusions from this first step warranted the combination of the Ice1-MP1 and Ice4-MP1 datasets resulting in both greater sample size and statistical power for further analyses.

-Based on the results of the Mann-Whitney U test, the second step involved using a series of Wilcoxon Signed-rank tests on the combined datasets in order to 1) assess the presence of statistical differences between the Controls and each Treatment L-DMSP/O and D-DMSP/O; 2) assess the potential effect of light on the concentrations and change rates of the reduced sulfur compounds under study (DMS, DMSPd

and DMSPp) by comparing paired dependent samples (repeated measures) from L-DMSP/O and D-DMSP/O. As recommended, df values are now provided for each statistical test performed.

- We deleted the section of the discussion where we tentatively estimated the overall size of the DMS reservoir in Arctic melt-ponds. We agree that a greater spatial coverage of MPs is needed to come up with a more robust estimate.

- In the initial submission, brine volumes were calculated from the T–S measurements. In the revised version, full depth temperature, salinity and brine volume profiles in sea ice are presented in a new figure (FigureÂǎ4) for stations Ice1, Ice3 and Ice4 in support of a more detailed discussion on the sea-ice surface permeability.

Author's response to specific comments:

R1: P2, L1: delete "natural" in first sentence of abstract, this word is not needed. Response: Done.

R1: P2, L12: This calculation of the DMS reservoir in Arctic melt-ponds is based on 2 single measurements of 2 very specific (=brackish) melt ponds in a very defined study area (e.g. the Canadian Archipelago). This reviewer considers up-scaling the results from this study to the entire Arctic as highly problematic. It is suggested to delete this estimate from the manuscript (see also page 16) or at least to delete this broad-brush estimate from the Abstract. Response: We deleted the calculation of the size of the DMS reservoir in Arctic FYI melt-ponds from the manuscript.

R1: P4, L17: No need to start a new paragraph. Response: This paragraph was merged with the previous paragraph.

R1: P4, L32: be more specific: "of melted ice samples" rather than "melt water samples" Response: The sentence now states "of melted ice [. . .]".

R1: P5, L1: The T and S data from the 10 cm surface ice allow the accurate calculation of the brine volume fraction according to established formulas, see e.g., Eicken, H.,

H. R. Krouse, D. Kadko, and D. K. Perovich, Tracer studies of pathways and rates of meltwater transport through Arctic summer sea ice, J. Geophys. Res., 107(C10), 8046, doi:10.1029/2000JC000583, 2002; an references therein. Applying these formulas (e.g. those for high T and low S sea ice values, e.g. Manninen and Leppaeranta 1988, cited in above reference), and using the values reported in the manuscript of T = -0.2C and S = 0 psu actually indicates "im"-permeable ice, while a T = -0.2C and S = 0.8 psu indicates a brine volume of about 20% (= highly permeable ice). This reviewer suggest that brine volumes are calculated for the T – S measurements and that a more detailed discussion on ice permeability/sea water percolation is given. Please also note that a) "the rule of 5s" is primarily based on a brine volume fraction of 5% (which can be achieved by different T-S combinations, including T = -5C and S = 5, b) that this percolation threshold is only valid during thermodynamic equilibrium, and c) also only applies for columnar ice (likely the case in these samples), but this surface ice might have also undergone some melting/metamorphosis). In summary this reviewers suggest a more detailed discussion of the sea ice permeability. The current conclusions are fine, but just stating "according to the rule of 5s" is insufficient. Response: A more detailed discussion on ice permeability/sea water percolation is now included in the manuscript. Full ice depth T and S profiles are now presented in figureÂǎ4 for stations Ice1, Ice3 and Ice4, and brine volumes were calculated from the T–S measurements. The method section was changed accordingly (P4, L28) and now states that : "In order to estimate the possibility of a connexion between the melt ponds sampled and the underlying sea ice (i.e. through ice permeability or water percolation), sea-ice salinity and temperature were measured. For each station where sea ice was sampled, an in situ sea-ice temperature profile was measured directly, at 0.1Âǎm intervals, using a high-precision thermometer (Testo 720). Corresponding sea-ice salinity profiles were also determined at 0.1Âǎm intervals. Each 0.1Âǎm section was cut with a handsaw, stored in a plastic container, and allowed to melt at room temperature. Bulk salinity of the melted ice section was determined using a conductivity probe (Cond 330i, WTW). Brine volume profiles were calculated using the recorded sea-ice bulk salinity and in

situ temperature (Cox and Weeks 1983, Petrich and Eicken 2010)".

Consequently, the result section was also modified (P9, L5). The text now reads: "Averaged values for bulk sea-ice salinity over the full thickness of the ice were 1.73, 2.83 and 3.75 at stations Ice1, Ice3 and Ice4, respectively. Maximum bulk salinity never exceeded 5.00 (Ice4, 1.2-1.3 m section). In situ temperatures, averaged over the full thickness of the ice, were -0.54 C, -0.52 C and -0.98 C at stations Ice1, Ice3 and Ice4, respectively, and reached a minimum value of -1.39 C (Ice4, 0.8-0.9 m section). Brine volume fraction constantly exceeded 10% in the ice profiles, except in the upper 0.1 m section of the Ice3 station."

R1: P5, L 11: "replicates" How many? Response: This was changed to "duplicates".

R1: P5, L25: This reviewer suggest to add a sentence and a definition of "HNA" here, e.g. what nucleic acid stain was used in this fly cytometry protocol? Response: The following sentence was added to the text: "Heterotrophic bacteria samples were stained with SYBR Green I and measured at 525 nm to quantify bacteria with Low Nucleic Acid (LNA; potentially less active) and High Nucleic Acid (HNA; potentially more active) content (Gasol and del Giorgio 2000, Lebaron et al. 2001). Analyses were performed on an Epics Altra flow cytometer (Beckman Coulter), fitted with a 488 nm laser (15 mW output; blue), using Expo32 v1.2b software (Beckman Coulter)."

R1: P6, L23: It is unusual to refer to PAR as "700-400", normally one would write "400-700". This also applies to the UVA and UVB wavelengths given in the text. Response: All the wavelengths presented in the text are now written in the suggested format.

R1: P 8, L25: As discussed above, this reviewer suggests to revisit the t-test statistics applied: It appears that N equals 2, which makes application of the t-test problematic. At least more explanation is needed. Response: As recommended, results from the Student's t-test are no longer presented. We nevertheless wanted to base our analysis on statistical tests. To do so, we explored the possibility of pooling our incubations data in order to increase 'n'. A non-parametric Mann-Whitney U test was first used to

determine whether the distributions of reduced-sulfur compounds (i.e. DMS, DMSPp and DMSPd) in the Ice1-MP1 and Ice4-MP1 incubations experiments were statistically different from one another. The difference in reduced-sulfur compound concentrations between the two incubation experiments was not found to be statistically significant (n=45, df=16 $\alpha$=0.05). As explained previously in the general comments section, this allows us to combine the results of Ice1-MP1 and Ice4-MP1 when testing for differences in responses between Treatments. This doubling of sample size (n) for each test (combining Ice1-MP1 and Ice4-MP1) led to an increase of the statistical power of the analysis conducted hereafter. A Wilcoxon Signed-rank test was used to assess potential statistical differences between the Controls and each Treatment L-DMSP/O and D-DMSP/O. Results reveal significant differences (p$\leq$0.05) between the Controls and each Treatment of the incubation experiments (n=30, df=8, $\alpha$=0.05). Further detail on the other statistical tests conducted is provided in the response to the "(P11, L10 -15)" comment.

R1: P 9, L3: This reviewer suggest to use the SI unit "m" rather than "cm" as unit for length measurements throughout the manuscript/figures. Response: "cm" was replaced with SI unit "m" throughout the manuscript and figures.

R1: P9, L7: As per above more details is required than just stating the "rule of fives". Response: More details are provided for this section in the response to the "P5, L1:" comment.

R1: P9, L 16: use singular, e.g. "detail" Response: The singular is now used in the text.

R1: P11, L10 -15: If "significantly" is used test-statistics should be given, also provide df value and/or N. Given the low N, these statistical results are of little relevance. Response: Each statement of the paragraph (between quotation marks) is now followed by a description of the statistical test used. As stated previously, results of Ice1-MP1 and Ice4-MP1 were combined (as justified by the the results of the Mann-Whitney U

test), resulting in an increase of the statistical power of the analysis conducted. "During both Ice1-MP1 and Ice4-MP1 incubation experiments, the light Treatment had no effect on the net changes in DMSPd concentrations between the L-DMSP/O and D-DMSP/O TreatmentsÂă[...]". →This was assessed using a Wilcoxon Signed-rank test (n=8, df=3, $\alpha$=0.05) comparing pairwise DMSPd concentrations at T6, T12, T18, and T24 for both incubation experiments Ice1-MP1 and Ice4-MP1, p$\geq$0.05 .

"[...] But significantly impacted the rates of net accumulation of DMS" → This was assessed using a Wilcoxon Signed-rank test (n=12, df=5, $\alpha$=0.05) with a significance level of p$\leq$0.05 comparing pairwise the DMS accumulation rates in L-DMSP/O versus D-DMSP/O at T0-T6, T6-T12, T12-T18, T18-T24, T6-T24 and daily rates (T0-T24) for both incubation experiments Ice1-MP1 and Ice4-MP1.

"The accumulation of DMS over 24h in the L-DMSP/O Treatments were consistently and significantly lower than in the corresponding D-DMSP/O Treatments (p$\leq$0.05) (Fig. 3b, d)." → This was assessed using a Wilcoxon Signed-rank test (n=8, df=3 ,$\alpha$=0.05) comparing pairwise DMS concentrations in L-DMSP/O versus D-DMSP/O at T6-T12-T18 and T24 in both incubation experiments Ice1-MP1 and Ice4-MP1.

R1: P 12, L 17: Sea "spray" rather than "spay" Response: Typo corrected.

AR1: P12, L28: Here "gravity drainage" and "brine flushing" are used to describe the same process, while classically "brine drainage" refers to the release of cold salt brines in surface-cooled sea ice, while "brine flushing" refers to the flushing out of salt through meltwater, e.g. they are technical terms used for different physical processes. Response: The technical terms are now correctly used in the text. The discussion on the salt movements through sea ice has also been amended. The corrected section is described below: (P12, L27) "It is also unlikely that sea-ice brine intrusion contributed to the salinization of the melt ponds since the ponded FYI sampled in this study appears to be almost fresh (using the terminology proposed in Vancoppenolle et al., 2007) (FigureÂă4). Consolidated cold FYI generally exhibits a characteristic C-shaped salinity

profile (Nakawo and Sinha, 1981) after loosing approximately two thirds of the initial seawater salt content through gravity drainage in winter (Kovacs, 1996). Then, according to the mushy-layer theoretical representation of sea ice, most of the salted brines are usually lost through full depth brine convection well before melt ponds start to form (Jardon et al., 2013). Finally, residual salts are lost during brine flushing events, typical of the summer season (Weeks and Ackley, 1986, Eicken et al., 2002; Vancoppenolle et al., 2007). The low salinity values and the flattened salinity profile observed in the sampled sea ice suggest that the ice had already been subjected to brine flushing. We thus exclude sea-ice brine enrichment of melt ponds as a significant salinization mechanism".

(P13, L3): "This leaves seawater intrusion through highly porous sea ice as the most likely process responsible for bringing salts, microorganisms, and DMS in melt ponds. Above a brine volume threshold of 5%, sea ice becomes permeable to fluid transport through its interconnected brine network (Golden et al., 1998). Melt ponds form and persist despite the high porosity of FYI due to the infiltration and subsequent freezing of a freshwater layer into the pore structure of sea ice that prevents percolation drainage of pond meltwater (Polashenski et al., 2017). Here, the brine volume fraction calculated for each 0.1 Ăm section always exceeds 10%, suggesting that sea ice was highly permeable throughout the full ice depth (except for the upper 0.1 Ăm of Ice3). As brines flushes out of the ice, seawater fills the channel network (Widell et al., 2006). Some degree of connectivity is thus expected to take place between superficial melt ponds and seawater. Specifying whether the intruding seawater originates from lateral or direct upward flow is difficult since these processes are not yet well understood (Vancoppenolle et al., 2007). Sea-ice freeboard was either low or negative near the melt ponds sampled (Table 1), suggesting that seawater intrusion through highly porous low-freeboard sea ice was possible in the observed sea ice. The somewhat higher freeboard measured at station Ice3 may indicate refreezing metamorphosis of snow. Sea-ice recrystallization could explain the impermeability of the upper 0.1 Ăm of sea ice at station Ice3. The low-freeboard configuration at stations Ice1 and Ice4 is

the general fate of melting sea ice, and inherent to the loss of sea-ice thickness. Our hypothesis of seawater intrusion through highly porous low-freeboard sea ice is also supported by the presence of both pelagic and ice-associated algae in the microbial assemblages of the melt ponds, along with the similarity observed between algal species composition in the waters of the melt ponds and those beneath the ice (Charette et al., personal comm.). The seeding of these seawater microorganisms into melt ponds may also affect the cycling of DMS as discussed in sect. 4.2".

P 13, L 10: No data are shown that demonstrate: "full depth desalinization" -> please clarify Response: This statement was removed from the manuscript. Calculations provided by Jardon et al. (2013) deal with the permeability threshold of sea ice with salinity greater than 5 psu. With a bulk sea ice salinity of 2.79 (averaged for the three stations), with a maximum value of 5.00 at station ice4 (1.2-1.3 m section), we fall outside of this range.

R1: P 13, L 20: Avoid the use of "significant" if no statistical test was conducted /or provide statistical results. Response: This was changed to "A daily net DMS production [...]".

Fig and Tables:

R1: Fig 3: Unusual numbering of panels: "c" should be "b" and "b" should be "c"? Response: The numbering was changed as suggested.

R1: Tab 7: "control" or "Control" -> consistency in spelling needed Response: The consistency of "control" spelling was checked and applied throughout the text.

Figure 4: $\breve{A}$ă In situ temperature $\breve{A}$ă(●)and bulk ice salinity (○) profiles of the sea ice surrounding the melt ponds sampled at stations Ice1 (a), Ice3 (b) and Ice4 (c). Temperature and salinity Âă values of each Âă10 cm sea ice section were used to calculate brine volumes (orange bars) throughout the full depth of sea ice, an indicator of sea ice permeability.

[Figure]

**Fig. 1.** In situ temperature (dark circles), bulk salinity (open circles) and brine volume (orange bars) profiles of the sea ice surrounding the melt ponds sampled at stations Ice1 (a), Ice3 (b) and Ice4 (c).

---

## Referee Comment (RC2) · G. Carnat (Referee) · 2 Feb 2018

General comments: The study of Gourdal et al. discuss the dynamics of the climate-active gas dimethylsulfide (DMS) in surface melt-ponds developing over Arctic first-year sea ice. The authors present an original data set of DMS(P) concentrations measured in nine melt-ponds combined with ancillary physical and biological parameters. Based on these data, the authors discuss several physical processes to explain the presence of DMS and microbial organisms in the melt-ponds. Then, the authors use incubations with stable isotope-labelled DMSP and DMSO to investigate de novo biological production of DMS in the melt-ponds via different pathways. As mentioned by the authors, this

study represents the first effort to characterize the cycling of DMS in Arctic melt-ponds, an interesting medium at the interface between sea ice and the atmosphere which importance is expected to increase in the future. Overall, the paper is well organized and well written. I would say that the methods regarding the DMS,P concentrations, incubations with isotopes, and ancillary biological parameters are adequate and well described. The DMS,P data, especially the results from the incubations experiments, are well presented and discussed in a very convincing way. That being said, I think that the physical component of the melt-pond/sea ice system is on the other hand poorly constrained in the study. There are numerous errors and approximation in each section of the manuscript regarding for instance sea ice permeability. I provided multiple suggestions and corrections in the specific comments detailed below and I strongly encourage the authors to follow these suggestions. This is my main criticism on the paper and I think this part should be improved before publication. I identified two other minor shortcomings. First, I think that the DMS cycling in melt-ponds could be better put in the general context of the DMS sea ice cycling, especially in the introduction. Second, I think that not enough precautions are taken when the regional estimates of the contribution of melt-ponds in the DMS cycle is assessed in the manuscript given the relative small number of samples considered. Also, this contribution should be compared to oceanic and sea ice contributions. Listed below are additional small and specific comments and recommendations. In summary, I suggest publication of the manuscript once the three (minor) issues identified above have been tackled and specific comments addressed. Specific comments: Please find a list of suggested references and reading at the end of the review. P2, L2 (and throughout the manuscript): "first-year" instead of "first year". P2, L3 (and throughout the manuscript): sea ice instead of sea-ice. Please be consistent throughout the manuscript. P2, L6: "In the Eastern Canadian Arctic", I would use "Canadian Arctic Archipelago" to be consistent with the title. P2, L7: Please check throughout the manuscript that "ca." is the proper scientific notation. Also, you could provide a range and standard deviation here between brackets. P2, L9: "Experiments conducted with" rather than "Results from experiments". This is a

little bit redundant with the next sentence. P2, L10: Bracket missing here. P2, L11-15: As explained in my general comments on the paper and on the conclusion, I think you should be a little bit more careful with this sentence since it is based on a very limited number of samples taken in a very limited area of the Arctic. While I believe it fits well in the conclusion where you have room to develop on limitations and future work to be conducted, you might want to remove it from the abstract. It is definitely not the key message of your paper. Should you keep it, I would at least put your estimate in perspective compared to other potential sources (open water, leads, sea ice itself,...). As it is, it is not clear for the reader if melt-ponds are a small or significant reservoir of DMS. P3, L12: "DMS-derived sulfate aerosols". P3, L14: Please indicate the two different backscattering effects of DMS-derived sulfate aerosols (direct and indirect through CCN). P3, L11-15: Please introduce here quickly the controversy about the CLAW hypothesis (cfr. e.g. Quinn and Bates, 2011, Green and Hatton, 2014) and the influence of DMS on a global scale. Then you can make the connection to the next sentence and talk about the influence of DMS on a more regional scale. P3, L16: "In remote pristine marine areas such as the polar regions". P3, L16: "Could be particularly important". P3, L19: Please add a reference here. P3, L20: The study of Rempillo et al. (2011) could also be cited here. P3, L22: This statement is not true. Please read again Stefels et al. (2007). The 95% mentioned refer to the fraction of DMS emitted from the ocean, not to the fraction of DMS in natural reduced sulfur emissions. I think a few other references (e.g. Lana et al., 2011, or the work of Bates) might be more appropriate. P3, L24: The reference is not correct. It should be Green and Hatton (2014). P3, L23-24: "Cellular metabolite" rather than "cellular compound", compound is a little bit vague. P3, L27: I would suggest to cite Lyon et al. (2016) for the osmoregulation, especially since you are talking about phytoplankton and not algae. Similarly, Karsten et al. (1996) seems appropriate for the cryoprotection hypothesis. P3, L32: in-situ. P3, L33: It would be nice to indicate in a short sentence how DMSP is released from the cell. P3, L24: Starting with "Between 1 and 40% of the DMSP...and ending page 3 line 9. The whole section is poorly structured and missing some important links. I would

suggest to rewrite following these lines: "...found in several phytoplankton species (DMSP particulate, or DMSPp) (see the review of Green and Hatton, 2014). DMSP plays several roles in phytoplankton, including osmoregulation (Lyon et al., 2016), cryoprotection (Karsten et al., 1996), and prevention of cellular oxidation (Sunda et al., 2002). Part of the DMSP produced by algae is released in the water column (dissolved DMSP, or DMSPd) where it is readily used by heterotrophic bacteria as carbon and sulfur sources (Kiene et al., 2000; Simo, 2001; Vila-Costa et al., 2006). The fraction of DMSPd consumed by heterotrophic bacteria and cleaved into DMS (DMS yield) may vary depending on the microbial community composition, its sulfur requirements, and the availability of other reduced forms of sulfur (Kiene et al., 2000; Stefels et al., 2007). DMSP-lyase enzymes are also present in several members of the microalgal groups Haptophyceae and Dinophyceae, and to a lesser extent Chrysophyceae (Niki et al., 2000). In addition to the DMSP cleavage pathway, a few studies have demonstrated the potential for reduction of dimethylsulfoxide (DMSO) by marine bacteria and phytoplankton as a source of DMS (e.g. Spiese et al., 2009; Asher et al., 2011). This metabolic pathway is however not ubiquitous among bacterial assemblages and may not be important quantitatively (Hatton et al., 2012; Green and Hatton, 2014). DMS concentrations in surface mixed layers are further influenced by three sinks: bacterial and photo-oxidation to DMSO, and ventilation to the atmosphere (Bates et al., 1994; Kieber et al., 1996; Simo and Pedros-Alio, 1999b; del Valle et al. 2007, 2009).Two regimes of ocean DMS production are documented. A "bloom-driven" regime in eutrophic regions where the DMS concentrations are controlled by phytoplankton blooms (Stefels et al., 2007), and a "stress-driven" regime in oligotrophic open ocean regions, where DMS concentrations are highly correlated to UV radiation (Toole and Siegel, 2004), nutrient limitation (Stefels, 2000), in-situ –temperatures (Karsten et al., 1996; van Rijssel and Gieskes, 2002), and –salinity (e.g. Kirst, 1996). Ultimately, between 1 and 40% of the DMSP produced by algae reaches the atmosphere as DMS (Simo and Pedros-Alio, 1999a)." P4, L5: It would be nice to write one or two sentences on particulate DMSO. P4, L10: As explained in my general comments, I think you need here

a paragraph on the importance of the sea ice ecosystem as a whole in the polar DMS cycle. This would help to better frame your study. It would be nice to introduce the important microbial biomass and DMS,P,O concentrations as well as the wide range of stresses encountered in the sea ice environment. Then you could talk about sea ice surface processes and introduce the cycling of DMS in melt-ponds. The review of Levasseur (2013) should help to put the melt-ponds in the general context of sea ice DMS production. Also, a few sentences on the specificity of the mobility of compounds within sea ice (i.e. permeability) should appear in this paragraph as it is a key part of your study. P4, L10-and further in the text. There is also some DMS melt-pond concentrations in the study of Leck and Persson (1996). This study should be cited in your publication. P4, L12: Please check that the DMSO reduction mentioned by Asher et al. (2011) was effectively detected in melt-ponds. If I remember correctly, the experiment was made in brine rather than in melt-ponds. High DMSO and DMS concentrations were indeed observed in melt-ponds but I believe the tracer experiment was exclusively made in brine, which is a very different medium. P4, L10-15: This is a little bit tricky. As you develop in the discussion section, the high DMS concentrations observed by Asher et al. (2011) were very likely related to the development of a surface ice community following flooding. I am fine with the fact that you develop this in the discussion section only, but I think you should already provide some hint in this introductory paragraph. It is a little bit misleading to only mention DMSO reduction and not to talk about the strong difference in microbial community development between the Arctic and Antarctic. P4, L15: "may also originate". Remove the also. You did not provide another explanation for the presence of DMS in the Arctic melt-ponds so far in the text. P4, L16: It would great to include here a few sentences on the typical environmental conditions/stress developing in surface melt-ponds, and how these conditions could influence DMS(P) production. P4, L17-18: It would be nice to rephrase and develop a little bit more this paragraph. The reader must be able to clearly identify the questions/gaps your study is going to address. For now it reads like the paper is just another data report...while I believe it is much more than that. Make it a little bit more appealing. P4, L25: You

could already indicate here between brackets (logistical constraints) why basic physical measurements were not conducted at Ice2. P4, L26: Please already define freeboard here. P4, L26: What motivated the sampling at a 3 m distance? Did you collect any other cores than the ones mentioned in this study? It would be nice to have an idea of the ice/snow thickness variability around the melt-ponds sampled. P4, L27: For sea ice physics discussions, it is always easier to measure salinity and temperature on the same ice core and at the same vertical resolution. It is always better to make full depth profiles as you will see later in my comments. P4, L28-29: Remove "According to a widely used protocol" and all the references that follow. Write: Sea ice temperature and bulk ice salinity were measured following Miller et al. (2015). Then: "Sea ice temperature was...". P4, L30: (and throughout the manuscript). Check for spacing between 5 and cm. I do not know what the recommendations of Biogeosciences are. P4, L28-31: Precision/accuracy of the probes should be indicated when available. Also check if you need to add trademark symbols next to the brands. P4, L32: "the bulk salinity of the melt aliquot". P4, L32: Permeability to fluid/gas transport is a more appropriate term than porosity here. P5, L1-3: and further in the discussion. Here you need to calculate the brine volume fraction in your sea ice samples following Leppäranta and Manninen (1988). The section needs to be rewritten. You cannot talk about permeability/porosity and the rule of fives without calculating and using the brine volume fraction. The rule of fives refers to three fives, salinity, temperature, and over all brine volume fraction. Temperature and salinity only are not sufficient to discuss permeability issues. Golden's research and all the research conducted on sea ice permeability and its influence on biogeochemistry (see Carnat et al. (2013), Carnat et al. (2014), Jardon et al. (2013), Zhou et al. (2013) indicate that sea ice becomes permeable to fluid transport when brine volume fraction reaches 5% (note that this threshold might vary substantially depending on ice texture for instance). The rule of fives stipulates that such a brine volume fraction (5%) corresponds for instance to a temperature of -5°C for an ice salinity of 5...not that the ice is permeable when the ice temperature is warmer than -5°C and the salinity higher than 5. P5, L7: Additional details are needed here.

It is not clear to me what the maximum pond fraction is. A picture of melt ponds has one and only one melt pond fraction. Regarding the mean, did you calculate it from multiple pictures? Could you provide the approximate area covered by the pictures? How many pictures were taken for each site? Did you try to assess the pond coverage digitally? Perhaps it would be great to indicate your estimated pond fraction for each sampling location in Fig1. P5, L11: How many replicates? It is not clear if chl a was measured on the ship or the filters stored. P5, L23-24: This is slightly confusing. Stored in liquid nitrogen (-196°C) or kept frozen at -80°C? P5, L27: Did you consider sampling multiple depths in the melt-ponds? Would you have expected homogeneity or a vertical gradient? Please quickly discuss this in the text. P5, L30: "to fill the glass serum bottles" remove the "the". P6, L11: Consider cutting in two sentences. "...into 5 ml FalconTM tube. DMSPd was quantified...". P6, L13: Please provide whenever possible an estimate of the error associated with every measurement. This is clearly missing for the measurement of DMS(P) concentrations. P6, L16: Dacey and Blough (1987) is perhaps a better reference here than Levasseur et al. (2006). P6, L26: "fresh water", do you mean milliQ water? Please specify. P6, L30: Consider using "duplicate" instead of "duplicated". P7, L10: This is I think the first time a Table is mentioned in the text. It should then be Table 1. I suggest to add a reference to Table 1 earlier in the text, in section 2.1. P8, L5: Is any fractionation expected during storage? P8, L6-10: Please provide the overall precision of the methods. P8, L9-10: This is very nice to read. P9, L5: Please add this 5 m information in the section 2.1 of the materials and methods part. P9, L6-8: Following my previous comments, this section needs to be rewritten. Also refrozen snow at the surface means superimposed ice, an ice texture known to be impermeable. This should be mention somewhere in the text. P9, L29: Please replace (see discussion) by "This will be discussed in section...". P11, L11: The use of "significantly" implies a statistical test which is not provided. P11, L17-30: You could make the paragraph a little bit lighter to read and easier to follow by removing some unnecessary instances of (m/z 68) and (m/z 62). P12, L1-2: See my previous comment. Please read the study of Leck and Persson (1996), cited in Levasseur (2013). There is also some

interesting work in glacial melt water ponds that you could consult and perhaps cite somewhere in the manuscript (De Mora et al., 1996), especially regarding to DMSO as a source of DMS. P12, L5: As stated before, I think this sentence is misleading and should be remove giving the fact that you provide further in the text a very plausible explanation for the difference. This explanation is moreover relatively logic for someone with a basic knowledge of sea ice biogeochemistry. P12, L14: What do you mean by "closed melt pond"? It seems that the melt-pond is exchanging material with seawater and the atmosphere. Please clarify. P12, L17 and 23: "Sea spray". P12, L27 – P13, L14: This whole section needs some rewriting. Full-depth gravity drainage should not be confused with flushing of surface melt-water. You should read a little bit more carefully the study of Jardon et al. (2013), but also Carnat et al. (2013) which describes the seasonal evolution of sea ice salinity (and brine salinity) in FYI in the Canadian Arctic (Amundsen Gulf, Beaufort Sea). Also, you definitely need to include brine volume fraction, Rayleigh number, and brine salinity here in the discussion. Unfortunately you only measured surface ice salinity and temperature, while full-depth profiles are generally necessary for this type of discussion. For instance, you could have 10 cm of sea ice with a low salinity due to percolating melt water with more saline layers underneath. Full-depth gravity drainage/convection requires both a connected brine network (sea ice permeable to fluid transport), and hence usually brine volumes above 5%, and an unstable brine density (brine salinity) profile. The combination of these two criteria can be expressed via a Rayleigh number. When sea ice warms up and reach the permeability threshold (expressed by the brine volume fraction, not the temperature), instability of the brine network (brine salinity being a direct function of sea ice temperature (Cox and Weeks (1983)), colder surface ice has saltier and denser brine than warmer bottom ice) can result in full-depth convection, brine being replaced by upward moving seawater. This usually occurs in mid-late spring (see the study of Carnat et al. (2013)) and results in some desalination of the ice cover (the upward moving seawater being less saline than the brine it is replacing). Following further warming in summer, surface melt water (melting snow or melting surface sea ice) percolates within the brine network leading to the process called flushing. This further decreases the bulk ice salinity down to values way under 2 psu as observed in your study. Warming will also dilute brine with pure ice melt water. I think that at the time of your sampling (based on the limited salinity and temperature data available), both full-depth gravity drainage and some flushing have already occurred. Hence, brine cannot indeed be responsible for the salinity observed in the melt-ponds. Now you still have to explain how to get seawater in contact with the melt-pond water through the porous brine network. Full-depth gravity drainage as suggested P13L10 makes no sense to me as the brine salinity do no support instability anymore. You also have to be a little bit careful with the use of the freeboard, especially citing Hudier et al. (1995). What Hudier et al. (1995) refers to is the loading of the sea ice surface with a significant amount of snow, depressing the surface sea ice level below the seawater level, leading to flooding of the ice surface, followed by gravity drainage. This is not really what you observed here. I agree that the decrease in sea ice thickness and development of the melt pond translate into a loss of freeboard, and that the melt-pond depth might approach the freeboard height, or even get below that height. Given the height of the freeboard and the depth of the melt-pond, seawater might infiltrate the porous ice texture via the brine network and start exchanging with the melt-pond. I am a little puzzled by the diffusion mechanism you suggest. It is probably true that at some point of the melt-pond evolution, infiltrated melt water might freeze and block the flushing of the pond by decreasing permeability in the ice layer under the melt-pond. No direct exchange with underlying seawater would then be possible. Diffusion could occur but would be a very slow process (especially through such layer), rather unlikely to explain the salinity change and biomass seeding observed in the pond. Alternatively, I wonder if the pond evolution could not alternate between phases of flushing, and phases of replenishment (pond depth being close to or below the freeboard height) with a mix of seawater and pond water. These phases would be controlled by small changes in ice temperature oscillating around the freezing temperature of the melt water. I think that the similarity in species composition between the melt-pond and under-ice seawater supports well this mechanism. P14,

L6-8: "over-flooding of sea ice". Replace by "flooding of the ice surface". Over-flooding is an odd term. P14, L6: Flooding could be better defined. P16, L7: Again, consider other data sets available. P16, L16: Modify "over-flooding". P16, L20: There are several studies providing direct (Nomura et al., (2012)) and indirect (Carnat et al., (2014)) evidences of DMS flux from FYI surface towards the atmosphere. P16, L24: These numbers should be put in perspective. How do they compare to the sea ice, ocean reservoirs? P16, L26: Is the average depth calculated from your data set or from literature observations? Your data set is relatively small. P16, L29: Wind velocity but also a better understanding of gas exchange between small fetch melt ponds and the atmosphere. References: Check the alphabetic order, Giamarelou et al. should be after Garrison. Figures and tables: Table 2: check the significant digits in the temperature values. Only physical characteristics are presented here, remove the chemical and biological characteristics from the caption. Table 7: Please be consistent with the significant digits. Figure 1: Please add a scale on figure 1b. As requested above, it would be nice to indicate the melt-pond fractions on each picture and an explanation of the calculation in the caption. Figure 3: Odd lettering of the figures.

Suggested references to read and/or add: -Quinn and Bates (2011). The case against climate regulation via oceanic phytoplankton sulphur emissions. Nature. -Green and Hatton (2014). The Claw hypothesis: a new perspective on the role of biogenic sulphur in the regulation of global climate. Oceanography and Marine Biology: An annual review. -Rempillo et al. (2011). Dimethyl sulfide air-sea fluxes and biogenic sulfur as a source of new aerosols in the Arctic fall. Journal of Geophysical Research. -Lana et al. (2011). An updated climatology of surface dimethylsulfide concentrations and emission fluxes in the global ocean. Global Biogeochemistry. -Lyon et al. (2016). Role of dimethylsulfoniopropionate as an osmoprotectant following gradual salinity shifts in the sea-ice diatom Fragilariopsis cylindrus. Environmental Chemistry. -Karsten et al. (1996). Dimethylsulfoniopropionate production in phototrophic organisms and its physiological function as a cryoprotectant. Biological and Environmental Chemistry of DMSP and Related Sulfonium Compounds. -Levasseur (2013). Impact of arctic meltdown on

the microbial cycling of sulphur. Nature Geosciences. -Leck and Persson (1996). The central Arctic Ocean as a source of dimethyl sulfide: seasonal variability in relation to biological activity. Tellus B. -Miller et al. (2015). Methods for biogeochemical studies of sea ice: The state of the art, caveats, and recommendations. Elementa: Science of the Anthropocene. -Lepparanta and Manninen (1988). The brine and gas content of sea ice with attention to low salinities and high temperatures. Finnish Institute of Marine Research Internal Report. -Carnat et al. (2013). Investigations on physical and textural properties of Arctic first-year sea ice in the Amundsen Gulf, Canada, November 2007 – June 2008 (IPY-CFL system study). Journal of Glaciology. -Carnat et al. (2014). Physical and biogeochemical controls on DMSP dynamics in ice shelf-influenced fast ice during a winter-spring and a spring-summer transitions. Journal of Geophysical Research-Oceans. -Zhou et al. (2013). Physical and biogeochemical properties in land-fast sea ice (Barrow, Alaska): Insights on brine and gas dynamics across seasons. Journal of Geophysical Research. -Dacey and Blough (1987). Hydroxide decomposition of dimethylsulfoniopropionate to form dimethylsulfide. Geophysical Research Letters. -De Mora et al. (1996). Aspects of the biogeochemistry of sulphur in glacial melt water ponds on the McMurdo Ice Shelf, Antarctica. Antarctic Science. -Cox and Weeks (1983). Equations for determining the gas and brine volumes in sea ice samples. Journal of Glaciology. -Nomura et al. (2012). Direct measurements of DMS flux from Antarctic fast sea ice to the atmosphere by a chamber technique. Journal of Geophysical Research Oceans.

---

## Author Comment (AC2) · 8 Mar 2018

Response to referee #2 in the interactive comments on "Dimethylsulfide dynamics in first-year sea ice melt ponds in the Canadian Arctic Archipelago"

Margaux Gourdal et al. margaux.gourdal@takuvik.ulaval.ca

General comments Referee G. Carnat: The study of Gourdal et al. discuss the dynamics of the climate-active gas dimethylsulfide (DMS) in surface melt-ponds developing over Arctic first-year sea ice. The authors present an original data set of DMS(P) concentrations measured in nine melt-ponds combined with ancillary physical and biological parameters. Based on these data, the authors discuss several physical processes to explain the presence of DMS and microbial organisms in the melt-ponds. Then, the authors use incubations with stable isotope-labelled DMSP and DMSO to investigate de novo biological production of DMS in the melt-ponds via different pathways. As mentioned by the authors, this study represents the first effort to characterize the cycling of DMS in Arctic melt-ponds, an interesting medium at the interface between sea ice and the atmosphere which importance is expected to increase in the future. Overall, the paper is well organized and well written. I would say that the methods regarding the DMS,P concentrations, incubations with isotopes, and ancillary biological parameters are adequate and well described. The DMS,P data, especially the results from the incubations experiments, are well presented and discussed in a very convincing way. That being said, I think that the physical component of the melt-pond/sea ice system is on the other hand poorly constrained in the study. There are numerous errors and approximation in each section of the manuscript regarding for instance sea ice permeability. I provided multiple suggestions and corrections in the specific comments detailed below and I strongly encourage the authors to follow these suggestions. This is my main criticism on the paper and I think this part should be improved before publication. I identified two other minor shortcomings. First, I think that the DMS cycling in melt-ponds could be better put in the general context of the DMS sea ice cycling, especially in the introduction. Second, I think that not enough precautions are taken when the regional estimates of the contribution of melt-ponds in the DMS cycle is assessed in the manuscript given the relative small number of samples considered. Also, this contribution should be compared to oceanic and sea ice contributions. Listed below are additional small and specific comments and recommendations. In summary, I suggest publication of the manuscript once the three (minor) issues identified above have been tackled and specific comments addressed.

Author's response to general comments: We thank the reviewer for his positive general evaluation of the paper and the insightful comments. The following general actions were taken: - The discussion on the physical component of the melt-pond/sea ice

system has been extensively amended following the suggestion of the reviewer. The discussion on sea ice permeability is now supported by brine volume calculations. We carefully considered the suggestion to use the Rayleigh number (Ra) in our analysis of ice permeability, however we finally decided to exclude this parameter. Details are provided in the supplementary material joined to our response to the reviewer. Briefly, we expect large uncertainty of Ra number calculation (VanCoppenolle et al., 2013) because brine loss during sampling of highly permeable sea ice, conditions encountered during our sampling period, has been shown to lead to an underestimation of bulk salinity (Notz et al., 2005) . - Several changes were made to the introduction section, as suggested by the reviewer. These changes include 1) a more detailed introduction of the DMS cycling (including more information on DMSO), 2) a clarified comparison with Antarctic melt ponds, and 3) a better description of the gap addressed by our study. -We deleted the estimate of the size of DMS reservoir in Arctic FYI melt ponds from the manuscript. We agree that small datasets, such as the one presented in our study, carry inherent limitations that make extrapolation calculations difficult.

Specific comments

P2, L2 (and throughout the manuscript): "first-year" instead of "first year". Response: This was corrected throughout the manuscript.

P2, L3 (and throughout the manuscript): sea ice instead of sea-ice. Please be consistent throughout the manuscript. Response: This was corrected throughout the manuscript.

P2, L6: "In the Eastern Canadian Arctic", I would use "Canadian Arctic Archipelago" to be consistent with the title. Response: We changed the sentence to fit the title formulation.

P2, L7: Please check throughout the manuscript that "ca." is the proper scientific notation. Also, you could provide a range and standard deviation here between brackets. Response: We did not find any Biogeosciences Discussions (BGD) guideline regarding the "ca." notation, but we checked other BGD papers. All of them used the tilde symbol "âĹij" and not "ca.". We now use the tilde symbol the manuscript. The text now reads as "[...] and increased linearly with salinity (rs = 0.84, p $\leq$ 0.05) from 2.6 up to 6.1 nmol l-1 (avg. 3.7 $\pm$Âǎ1.6 nmol l-1) in brackish melt ponds."

P2, L9: "Experiments conducted with" rather than "Results from experiments". This is a little bit redundant with the next sentence. Response: The sentence starts now with "Experiments conducted with [...]".

P2, L10: Bracket missing here. Response: A bracket was added.

P2, L11-15: As explained in my general comments on the paper and on the conclusion, I think you should be a little bit more careful with this sentence since it is based on a very limited number of samples taken in a very limited area of the Arctic. While I believe it fits well in the conclusion where you have room to develop on limitations and future work to be conducted, you might want to remove it from the abstract. It is definitely not the key message of your paper. Should you keep it, I would at least put your estimate in perspective compared to other potential sources (open water, leads, sea ice itself, ...).As it is, it is not clear for the reader if melt-ponds are a small or significant reservoir of DMS. Response: We agree that small datasets such as presented in our study carry inherent limitations that make extrapolation calculations difficult. We therefore decided to delete the estimate of the size of DMS reservoir in Arctic FYI melt ponds from the manuscript.

P3, L12: "DMS-derived sulfate aerosols". Response: The word "sulfate" was added to the text.

P3, L14: Please indicate the two different backscattering effects of DMS-derived sulfate aerosols (direct and indirect through CCN). Response: Please see the re-written paragraph in the P3, L11-15 response below.

P3, L11-15: Please introduce here quickly the controversy about the CLAW hypothesis

(cfr. e.g. Quinn and Bates, 2011, Green and Hatton, 2014) and the influence of DMS on a global scale. Then you can make the connection to the next sentence and talk about the influence of DMS on a more regional scale. Response: The paragraph starting (P3, L11) was modified according to the suggestions P3, L12 ; P3, L14 ; P3, L11-15 ; P3, L16 ; P3, L19 ; P3, L20: The paragraph now reads: "Dimethylsulfide (DMS) is the main natural source of reduced sulfur for the atmosphere (Bates et al., 1992). Between 17.6 to 34.4 Tg of sulfur are released annually from the ocean to the atmosphere (Lana et al., 2011), accounting for 50-60% of the natural reduced sulfur emitted (Stefels et al., 2007). DMS is also a climate-relevant gas potentially involved in a feedback loop known as the "CLAW" hypothesis (Charlson et al., 1987) linking biology and climate through the production of DMS-derived sulfate aerosols. According to CLAW, DMS emissions may affect the global radiation budget directly through the scattering of incoming solar radiation, and indirectly via the production of cloud condensation nuclei (CCN) leading to the genesis of longer-lived clouds with higher albedo (Twomey, 1974; Albrecht, 1989). Inspiring three decades of research and hundreds of publications, the feedback mechanism proposed by Charlson et al. (1984) remains yet to be demonstrated in its entirety (e.g. Ayers and Cainey, 2008 ). Although modelling results show that DMS emissions may have a negative radiative effect (e.g. Bopp et al., 2004; Gunson et al., 2006; Thomas et al., 2010), CCN may exhibit a low sensitivity to changes in DMS on a global scale (Woodhouse et al., 2010). Recent studies questioning the relative importance of DMS in new particle formation have emerged, suggesting that the global CLAW feedback may be weak (e.g. Quinn and Bates, 2011; Green and Hatton, 2014). On a regional scale however, the response of CCN production to change in DMS may vary by a factor of 20 (Woodhouse et al., 2010). The impact of DMS emissions on cloud properties (through the production of CCN) could be particularly important in remote pristine marine areas such as the polar regions (Carslaw et al., 2013). In the Southern Ocean, DMS may have contributed up to 33% of the increase in CCN observed south of 65°S as a response of increased wind speed since the early 1980s (Korhonen et al., 2010). The summertime Arctic marine

boundary layer (MBL) is left relatively clean after seasonal wet deposition of particles and reduced atmospheric transport of aerosols from anthropogenic sources at lower latitudes (Stohl, 2006; Browse et al., 2012; Croft et al., 2016). Such pristine conditions, combined with thermally stable MBL are typical of the Arctic summertime (e.g. Aliabadi et al., 2016). Clean Arctic air masses allow ultrafine (5 - 20 nm diameter) particle formation (Burkart et al., 2016), and the potential growth of secondary marine organic aerosols (including DMS-derived particles) into cloud condensation nuclei (Willis et al., 2016). Hence, the Arctic is a favourable terrain for new particle formation from biogenic DMS (Chang et al., 2011; Rempillo et al., 2011; Collins et al., 2017; Giamarelou et al., 2016; Mungall et al., 2016; Willis et al., 2016)."

P3, L16: "Could be particularly important" "In remote pristine marine areas such as the polar regions". Response: "Could be" and "Such as the polar regions" were added in the text (please sea answer to P3, L11-15 above).

P3, L19: Please add a reference here. Response: References were added to the text (please sea answer to P3, L11-15 above). The sentence now reads: "[...] Such pristine conditions, combined with thermally stable MBL are typical of the Arctic summertime (e.g. Aliabadi et al., 2016). Clean Arctic air masses allow ultrafine (5 - 20 nm diameter) particle formation (Burkart et al., 2016), and the potential growth of secondary marine organic aerosols (including DMS-derived particles) into CCN (Willis et al., 2016)."

P3, L20: The study of Rempillo et al. (2011) could also be cited here. Response: This reference was added to the text (see response to P3, L11-15 above).

P3, L22: This statement is not true. Please read again Stefels et al. (2007). The 95% mentioned refer to the fraction of DMS emitted from the ocean, not to the fraction of DMS in natural reduced sulfur emissions. I think a few other references (e.g. Lana et al., 2011, or the work of Bates) might be more appropriate. Response: We corrected this statement and added the suggested references. The new sentence reads: "Between 17.6 to 34.4 Tg of sulfur are released annually from the ocean to the atmosphere

(Lana et al., 2011), accounting for 50-60% of the natural reduced sulfur emitted (Stefels et al., 2007).".

P3, L24: The reference is not correct. It should be Green and Hatton (2014) Response: The author's names were corrected.

P3, L23-24: "Cellular metabolite" rather than "cellular compound", compound is a little bit vague. Response: We use the formulation "metabolite" as suggested.

P3, L27: I would suggest to cite Lyon et al. (2016) for the osmoregulation, especially since you are talking about phytoplankton and not algae. Similarly, Karsten et al. (1996) seems appropriate for the cryoprotection hypothesis. Response: The suggested references were added in the text.

P3, L32: in-situ. Response: "In situ" was written following BGD guidelines: "Common Latin phrases are not italicized (for example, et al., cf., e.g., a priori, in situ, bremsstrahlung, and eigenvalueound)".

P3, L33: It would be nice to indicate in a short sentence how DMSP is released from the cell. Response: Please see the additional information in the response to comment P3, L24.

P3, L24: Starting with "Between 1 and 40% of the DMSP...and ending page 3 line 9. The whole section is poorly structured and missing some important links. I would suggest to rewrite following these lines: "...found in several phytoplankton species (DMSP particulate, or DMSPp) (see the review of Green and Hatton, 2014). DMSP plays several roles in phytoplankton, including osmoregulation (Lyon et al., 2016), cryoprotection (Karsten et al., 1996), and prevention of cellular oxidation (Sunda et al.,2002). Part of the DMSP produced by algae is released in the water column (dissolved DMSP, or DMSPd) where it is readily used by heterotrophic bacteria as carbon and sulfur sources (Kiene et al., 2000; Simó, 2001; Vila-Costa et al., 2006). The fraction of DMSPd consumed by heterotrophic bacteria and cleaved into DMS (DMS yield) may vary depend-

ing on the microbial community composition, its sulfur requirements, and the availability of other reduced forms of sulfur (Kiene et al., 2000; Stefels et al., 2007). DMSP-lyase enzymes are also present in several members of the microalgal groups Haptophyceae and Dinophyceae, and to a lesser extent Chrysophyceae (Niki et al., 2000). In addition to the DMSP cleavage pathway, a few studies have demonstrated the potential for reduction of dimethylsulfoxide (DMSO) by marine bacteria and phytoplankton (e.g. Spiese et al., 2009; Asher et al., 2011), and non-marine Antarctic shelf ponds bacteria (De Mora et al., 1996) as sources of DMS. This metabolic pathway is however not ubiquitous among bacterial assemblages (Hatton et al., 2004 ; Green and Hatton, 2014). DMS concentrations in surface mixed layers are further influenced by three sinks: bacterial and photo-oxidation to DMSO, and ventilation to the atmosphere (Bates et al., 1994; Kieber et al., 1996; Simó and Pedros-Alio, 1999b; del Valle et al. 2007, 2009). Two regimes of ocean DMS production are documented. A "bloom-driven" regime in eutrophic regions where the DMS concentrations are controlled by phytoplankton blooms (Stefels et al., 2007), and a "stress-driven" regime in oligotrophic open ocean regions, where DMS concentrations are highly correlated to UV radiation (Toole and Siegel, 2004), nutrient limitation (Stefels, 2000), in-situ –temperatures (Karsten et al., 1996; van Rijssel and Gieskes, 2002), and –salinity (e.g. Kirst, 1996). Ultimately, between 1 and 40% of the DMSP produced by algae reaches the atmosphere as DMS (Simó and Pedros-Alio, 1999a).". Response: Sentences of the original paragraph were rearranged as suggested, with some modifications to facilitate the transition between paragraphs. The new proposed paragraph contains additional information regarding particulate DMSO (as recommended in specific comment P4, L5) and mechanisms for DMSP release from the cell (as recommended in specific comment P3, L330). New paragraph: "DMS stems mainly from the enzymatic cleavage of dimethylsulfoniopropionate (DMSP) by algal and bacterial DMSP-lyases. DMSP is a cellular metabolite found in several phytoplankton species as particulate DMSP (DMSPp) (see the review of Green and Hatton, 2014). DMSPp plays various roles in phytoplankton, including osmoregulation (Lyon et al., 2016), cryoprotection (Karsten et al., 1996), and prevention of cellular oxidation (Sunda et al., 2002). Part of the DMSPp produced by algae is released in the water column as dissolved DMSP (DMSPd) via several pathways reviewed in Stefels et al. (2007), including active exudation, cell lysis, viral lysis and zooplankton grazing. DMSPd is then readily available to heterotrophic bacteria as carbon and sulfur sources (Kiene et al., 2000; Simo, 2001; Vila-Costa et al., 2006). The fraction of DMSPd consumed by heterotrophic bacteria and enzymatically cleaved by DMSP-lyases into DMS (DMS yield) may vary depending on the composition of microbial communities, their sulfur requirements, and the availability of other reduced forms of sulfur (Kiene et al., 2000; Stefels et al., 2007). DMSP-lyase enzymes are also present in several members of the microalgal groups Haptophyceae and Dinophyceae, and to a lesser extent Chrysophyceae (Niki et al., 2000). Ultimately, between ∼1 and 40% of the DMSP produced by algae reaches the atmosphere as DMS (Stefels et al., 2007; Simo and Pedros-Alio, 1999a).".

P4, L5: It would be nice to write one or two sentences on particulate DMSO. Response: The following sentences were added: "In addition to the DMSP enzymatic cleavage pathway, DMS production may arise from dimethylsulfoxide (DMSO) reduction by various groups of marine bacteria including proteobacteria (e.g. Vogt et al., 1997), members of the Roseobacter group (Gonzalez et al., 1999) and mat-forming cyanobacteria (van Bergeijk and Stall., 1996). However, the ubiquity of this DMSO-to-DMS reduction pathway amongst bacterial assemblages has not been established (Hatton et al., 2012). A limited number of phytoplankton species could also be involved in the reduction of DMSO into DMS (e.g. Fuse et al., 1995; Spiese et al., 2009). Increasing evidence suggests that particulate DMSO (DMSOp) may be directly synthesized by a potentially wide range of marine phytoplankton (e.g. Lee and de Mora, 1996) and could be involved in osmoprotection, cryoprotection (Lee and de Mora 1999), and antioxidant protective mechanisms (Sunda et al., 2002). As for dissolved DMSO (DMSOd), it is ubiquitous in seawater and continuous improvements in analytical techniques suggest that DMSOd may be as abundant as DMS in surface waters (e.g. Simo et al., 2000). DMSO is also a known sink for DMS (Hatton et al., 2004) via bacterial and

photo oxidation of DMS to DMSO. Vertical Mixing and ventilation are also major removal processes influencing DMS concentrations in surface mixed layers (Bates et al., 1994; Kieber et al., 1996; Simó and Pedrós-Alió, 1999b; del Valle et al. 2007, 2009).".

P4, L10: As explained in my general comments, I think you need here a paragraph on the importance of the sea ice ecosystem as a whole in the polar DMS cycle. This would help to better frame your study. It would be nice to introduce the important microbial biomass and DMS,P,O concentrations as well as the wide range of stresses encountered in the sea ice environment. Then you could talk about sea ice surface processes and introduce the cycling of DMS in melt-ponds. The review of Levasseur (2013) should help to put the melt-ponds in the general context of sea ice DMS production. Response: A new paragraph was written as follows: "Ice-associated environments such as bottom sea ice, brine channels, melt ponds, under-ice surface waters, and leads provide complex and dynamic habitats to diverse microorganism communities involved in sulfur cycling (Levasseur, 2013). In the Arctic, the highest microalgal biomasses are found in the bottom ∼0.1 m of sea ice, with Chlorophyll a (Chl a) concentrations several orders of magnitude above values for under-ice waters values (e.g. Legendre et al. 1992). A similar pattern of DMSP, DMSO and DMS build-up in bottom ice has been reported both in the Arctic and Antarctica (Kirst et al., 1991; Levasseur et al. 1994; Turner et al., 1995; DiTullio et al., 1995; Lee et al., 2001; Trevena et al., 2003; Trevena and Jones 2006; Delille et al., 2007; Tison et al., 2010; Asher et al., 2011; Nomura et al., 2012; Galindo et al., 2015). For example, DMSPp concentrations up to 15 000 nmol l–1 have been documented during spring in bottom FYI of the Eastern Arctic (Galindo et al., 2014). DMSP, DMSO and DMS are also present throughout the ice column within the brine network (Levasseur et al., 1994; Trevena and Jones, 2006; Asher et al., 2011). Given that primary producers are the sole source of DMSP, very high ice concentrations of Chl a are often correlated with DMSP through a first order relationship (Levasseur, 2013). Beyond inter-specific differences in DMSP cellular contents (e.g. Keller et al., 1989; Stefels et al. 2004), environmental forcings are known to control DMSP, DMSO and DMS concentrations. In ice-associated environments, brine

volume fraction might also be key in explaining DMS cycling variability via the control of ice permeability (Carnat et al., 2014). The melting season is a pivotal and productive period for these sulfur-containing compounds. Structural changes within sea ice during the melt season, namely increases in brine volume fraction and ice desalination, result in increased connectivity and permeability in the warming sea ice (Willis et al., 2006; Polashenski et al., 2012) and influence DMSP and DMS cycling (Carnat et al., 2014). Also, phytoplankton blooms developing under the ice during the melting period have been shown to produce large quantities of DMSPp, potentially leading to a build-up of DMS concentrations (Levasseur et al., 1994). In spite of the spatial importance of melt ponds, only few studies have investigated their role as a source of DMS for the Arctic atmosphere (e.g. Levasseur, 2013; Nomura et al., 2012).".

P4, L10-and further in the text. There is also some DMS melt-pond concentrations in the study of Leck and Persson (1996). This study should be cited in your publication. Response: Leck and Persson (1996) reference was added to the manuscript. The authors mention "negligible levels of DMS" in the "samples collected in the melt ponds [...]"encountered in "an area of multiple year ice". The text was modified as follows: "Four studies have specifically reported on DMS in melt ponds so far. They reveal negligible DMS concentrations in MYI ice melt ponds in the Central Arctic Ocean, and concentrations up to 2.2 nmol l-1 in the High Arctic (Sharma et al., 1999). In Antarctica, DMS concentrations ranging between 1.1 and 3.7 nmol l-1 and between below the detection limit (d.l.) and 250 nmol l-1 were measured in two studies (Nomura et al., 2012 and Asher et al. 2011, respectively).".

P4, L12: Please check that the DMSO reduction mentioned by Asher et al. (2011) was effectively detected in melt-ponds. If I remember correctly, the experiment was made in brine rather than in melt-ponds. High DMSO and DMS concentrations were indeed observed in melt-ponds but I believe the tracer experiment was exclusively made in brine, which is a very different medium. Response: Indeed, Asher et al. (2011) report results from experiments made in brine (as well as in slush and surface open waters), but not

in melt ponds. The authors suggest that high DMS/P/O concentrations measured in melt ponds might be associated with rapid DMS/P/O cycling and DMSO reduction, but future work is needed to firmly conclude. We clarified this in the amended version of the manuscript as follows: "In the latter study, bacterial DMSO reduction was suggested as a possible mechanism responsible for the high DMS concentrations observed although no actual rates of DMS production, either from DMSO or DMSP, were measured.".

P4, L10-15: This is a little bit tricky. As you develop in the discussion section, the high DMS concentrations observed by Asher et al. (2011) were very likely related to the development of a surface ice community following flooding. I am fine with the fact that you develop this in the discussion section only, but I think you should already provide some hint in this introductory paragraph. It is a little bit misleading to only mention DMSO reduction and not to talk about the strong difference in microbial community development between the Arctic and Antarctic. Response: The difference in surface communities between Arctic and Antarctic ecosystems, and the prevalence of surface ice communities following flooding was introduced as follows: "High DMS concentrations reported in the Antarctic are most likely related to the development of a surface ice community following flooding. Several studies document melt pond colonization by micro-, nano- and pico-sized algae as well as bacteria (Bursa, 1963; Gradinger et al., 2005; Elliott et al., 2015), suggesting that DMS in melt ponds may originate from algal and bacterial metabolism.". Additional focus on the Antarctic versus Arctic sea ice dynamics are also provided in the discussion section (sect. 4.2.2). The text in the 4.4.2 section now states: "Extremely high gross DMS production rates from DMSO reduction, up to $105 \pm 24$ nmol l- 1 d- 1, were measured within Antarctic sea ice brines by Asher et al. (2011). The authors suggested that this mechanism could also potentially be responsible for the high DMS concentrations (up to 250 nmol l-1) measured in Antarctic melt ponds. The absence of DMS production from 13C-DMSO in the melt ponds studied here may then reflect potential differences in microbial assemblages within melt ponds, as the metabolic ability to convert DMSO into DMS is not ubiquitous among bacterial communities (Hatton et al., 2012; Hughes et al., 2014). In support of this hypothesis, it

has been shown that between 70 and 78% of the operational taxonomic units (OTU), a marker of microbial diversity, in Arctic and Southern Ocean surface water communities are unique to their region (Ghiglione et al., 2012). Observed differences in the biological characteristics of melt ponds between the poles could also reflect divergent sea ice dynamics. Antarctic sea ice salinity is higher by 0.5 to 1.0% than in Arctic sea ice (Gow et al., 1982, 1987) and the C-shaped salinity profile that is typical in fully formed Arctic FYI is not as prominent in Southern Ocean sea ice (Eicken, 1992). Antarctic sea ice is commonly subjected to intense rafting. Flooding, a process whereby heavy snow load pushes the ice below the water level, is common in the Antarctic and results in the formation of snow ice (Hunke et al., 2011). Antarctic melt ponds studied in Asher et al. (2011) may have been subjected to this flooding leading to the formation of salted "freeboard layers" (Haas et al., 2001; Massom et al., 2006). This is supported by the reported highest salinities in the top sea ice layers and the subsequent salinity decrease throughout the ice profile (Asher et al. 2011). Such configuration may bring highly productive microbial communities at the surface of the ice, potentially responsible for the high DMS concentrations observed in melt ponds. [...].".

P4, L15: "may also originate". Remove the also. You did not provide another explanation for the presence of DMS in the Arctic melt-ponds so far in the text. Response: The word "also" was removed.

P4, L16: It would great to include here a few sentences on the typical environmental conditions/stress developing in surface melt-ponds, and how these conditions could influence DMS(P) production. Response: We agree that introducing the incidence of known plankton stressors such as elevated light, substrate limitation and osmotic shock would make valuable additions. However, the length of the introduction has already considerably increased in the revised version of the manuscript. We therefore decided to not add this additional information to the introduction. We mention the typical environmental stress in melt ponds in several instances throughout the manuscript (e.g. In the introduction: "environmental forcings are known to control DMSP, DMSO and DMS

concentrations" and in section 4.2.3 "Fast and transient intracellular accumulation of compatible solutes, such as DMSP, may serve as an adaptive strategy by microbial cells to help cope with fluctuations of the surrounding environment, increasing their tolerance to osmotic and thermal stresses for example (Welsh, 2000).").

P4, L17-18: It would be nice to rephrase and develop a little bit more this paragraph. The reader must be able to clearly identify the questions/gaps your study is going to address. For now it reads like the paper is just another data report ...while I believe it is much more than that. Make it a little bit more appealing. Response: The rewritten paragraph reads as follow: "Considerable efforts have been dedicated to the understanding of underlying process controlling the physics of melt ponds and their feedbacks on climate through the control of surface energy balance of the ice (Lüthje et al., 2006; Polashenski et al., 2017). However, little is known about their biogeochemistry. Four studies have specifically reported on DMS in melt ponds so far. They reveal negligible DMS concentrations in MYI ice melt ponds in the Central Arctic Ocean, and concentrations up to 2.2 nmol l-1 in the High Arctic (Sharma et al., 1999). In Antarctica, DMS concentrations ranging between 1.1 and 3.7 nmol l-1 and between below the detection limit (d.l.) and 250 nmol l-1 were measured in two studies (Nomura et al., 2012 and Asher et al. 2011, respectively). In the latter study, bacterial DMSO reduction was suggested as a possible mechanism responsible for the high DMS concentrations observed although no actual rates of DMS production, either from DMSO or DMSP, were measured. High DMS concentrations reported in the Antarctic are most likely related to the development of a surface ice community following flooding. Several studies document melt pond colonization by micro-, nano- and pico-sized algae as well as bacteria (Bursa, 1963; Gradinger et al., 2005; Elliott et al., 2015), suggesting that DMS in melt ponds may originate from algal and bacterial metabolism. Yet, in situ DMS production had never been measured nor had key mechanisms been identified. Here, we report on the DMS concentrations in nine melt ponds located in the Eastern Canadian Arctic Archipelago (CAA), and on the prerequisites and processes responsible for the presence of this climate-active gas. This is the first attempt to assess the dynamics of DMS

in Arctic melt ponds. We identified sea ice permeability as a major control of DMS production in melt ponds, mediating the transport of both DMS and DMS-producing communities toward the surface of sea ice. We also provide the first evidence for direct in situ DMS production in Arctic melt ponds. We propose that seasonally melting sea ice might become increasingly prone to DMS production as FYI become largely predominant at the regional scale.".

P4, L25: You could already indicate here between brackets (logistical constraints) why basic physical measurements were not conducted at Ice2. Response: "(logistical constraints associated with the ship time line)" was added in the first sentence of the paragraph.

P4, L26: Please already define freeboard here. Response: We added a definition of freeboard in the text: "[...] freeboard (the height of sea ice above the ocean surface), [...].".

P4, L26: What motivated the sampling at a 3 m distance? Did you collect any other cores than the ones mentioned in this study? It would be nice to have an idea of the ice/snow thickness variability around the melt-ponds sampled. Response: The text now states : "The 3 m distance was a compromise between maximizing the proximity of ice and melt pond samples and minimizing melt pond disturbance during sampling operations. Ice and freeboard thickness presented in table 1 are averaged values of the 7 (Ice1) to 8 (Ice3 and Ice4) ice cores sampled at each station between the team members for their respective projects." We aslo modified the caption accordingly: "Table 1: Physical characteristics of the sea ice surrounding the melt ponds. Note that only melt pond sampling (i.e. no ice sampling) was conducted at station Ice2 due to ship-related logistical constraints. A negative freeboard height indicates that the ice surface was locally below the mean sea level. n/a stands for non-available data. Ice thickness and freeboard values are averages of 7 (Ice1) to 8 (Ice3 and Ice4) ice cores sampled at each station.".

P4, L27: For sea ice physics discussions, it is always easier to measure salinity and temperature on the same ice core and at the same vertical resolution. It is always better to make full depth profiles as you will see later in my comments. Response: Full ice depth temperatures and salinity profiles are now presented in a figure (new figure 3) and included in the discussion. Please see response to comment P 12, L27 – P13, L14 for further details.

P4, L28-29: Remove "According to a widely used protocol" and all the references that follow. Write: Sea ice temperature and bulk ice salinity were measured following Miller et al. (2015). Then: "Sea ice temperature was ...". Response: The phrase 'According to a widely...' was removed and the reference Miller et al (2015) was added.

P4, L30: (and throughout the manuscript). Check for spacing between 5 and cm. I do not know what the recommendations of Biogeosciences are. Response: Biogeosciences recommends "not to hyphenate modifiers containing abbreviated units (e.g. "3-m stick" should be "3 m stick")". As we did not find any other recommendations, we inserted non-breaking spaces throughout the manuscript to avoid line breaking between numbers and their units.

P4, L28-31: Precision/accuracy of the probes should be indicated when available. Also check if you need to add trademark symbols next to the brands. Response: Precision of the probes were indicated in the amended manuscript. Trademarks/registered symbols were revised throughout the text.

P4, L32: "the bulk salinity of the melt aliquot". Response: As suggested by the Anonymous Referee #1, we changed this to "Bulk salinity of the melted ice section".

P4, L32: Permeability to fluid/gas transport is a more appropriate term than porosity here. Response: The sentence was changed to: "Permeability to fluid transport was assessed with brine volume profile calculations from bulk salinities and sea ice temperatures following equations from Leppäranta and Manninen (1988) for sea ice temperatures > -2°C (Fig. 3).".

P5, L1-3: and further in the discussion. Here you need to calculate the brine volume fraction in your sea ice samples following Leppäranta and Manninen (1988). The section needs to be rewritten. You cannot talk about permeability/porosity and the rule of fives without calculating and using the brine volume fraction. The rule of fives refers to three fives, salinity, temperature, and over all brine volume fraction. Temperature and salinity only are not sufficient to discuss permeability issues. Golden's research and all the research conducted on sea ice permeability and its influence on biogeochemistry (see Carnat et al. (2013), Carnat et al. (2014), Jardon et al. (2013), Zhou et al. (2013) indicate that sea ice becomes permeable to fluid transport when brine volume fraction reaches 5% (note that this threshold might vary substantially depending on ice texture for instance). The rule of fives stipulates that such a brine volume fraction (5%) corresponds for instance to a temperature of -5◦C for an ice salinity of 5...not that the ice is permeable when the ice temperature is warmer than -5◦C and the salinity higher than 5. Response: Full ice depth T and S profiles are now presented in a new figure (Fig. 3) for stations Ice1, Ice3 and Ice4. Corresponding brine volume profiles were calculated using the recorded sea ice bulk salinity and in situ temperature (Leppäranta and Manninen, 1988). Calculated brine volume fraction constantly exceeded 10% in the ice profiles, except in the upper 0.1 m section of the Ice3 station. For the latter, we likely observed the effects of refreezing metamorphosis of snow and / or sea ice recrystallization. As mentioned in Polashenski et al. (2017) after Golden et al. (1998) and Golden (2003), liquid inclusions in columnar sea ice become interconnected once brine volume fraction reaches 5% in columnar FYI and 10% in granular FYI. Although no ice structure analysis was conducted during our study, columnar ice is expected to dominate FYI stratigraphy in the Arctic (Thomas and Dieckmann, 2008). We therefore decided to use the 5% brine volume threshold for the ice permeability in our study. These additional data show that the sea ice was (with the exception of the upper 0.1 m in Ice3 ice core) highly permeable throughout the three full ice profiles.

P5, L7: Additional details are needed here. It is not clear to me what the maximum pond fraction is. A picture of melt ponds has one and only one melt pond fraction.

[Figure]

Regarding the mean, did you calculate it from multiple pictures? Could you provide the approximate area covered by the pictures? How many pictures were taken for each site? Did you try to assess the pond coverage digitally? Perhaps it would be great to indicate your estimated pond fraction for each sampling location in Fig1. Response: We did not assess the melt pond fraction (MPF) digitally. Although we agree that a digital assessment of MPF would represent valuable information, detection and quantification of sea ice surfaces, including melt ponds coverage, is still an ongoing research field (Scharien et al., 2017 ; Wright and Polashenski, submitted). Impact of melt ponds on the ice albedo (e.g. Flocco et al., 2012), and the link between spring melt pond fraction and September sea ice minimum extent (Schröder et al., 2014), both using MPF, are still actively explored. Various techniques including low-level aerial photographs (e.g. Derksen et al., 1997), satellite based passive microwave observations (Fetterer and Untersteiner, 1998), synthetic aperture radar (Yackel and Barber, 2000), Moderate Resolution Imaging Spectroradiometer imagery (MODIS) (e.g. Tschudi et al., 2008) and LiDAR (Light Detection and Ranging) (Landy et al., 2014) are used throughout the literature. These techniques are beyond the scope of the present study. In addition, MODIS data of melt pond fraction after 2011 are not publicly available yet. Also, the use of melt pond coverage data is minimal in our study. That being said, the term "maximum" was removed from the text as it infers that there was also a minimum pond fraction at each station. Between two and three persons documented the sampling operations by taking 5 to 10 digital photographs using a hand-held camera from the bridge (17 m height) during each station. Their individual assessments were then compared and averaged values are presented. An approximative size scale and the estimated MPF originally presented in Table 1 were added to figure 1 and removed from Table 1.

P5, L11: How many replicates? It is not clear if chl a was measured on the ship or the filters stored. Response: Measurements were done in duplicate. This was changed in the manuscript. Chl a measurements were conducted on board. The text now states: "[...] duplicates of in situ pond water were filtered onto Whatman GF/F 25 mm filters. Pigments were extracted in 90% acetone for 18 to 24 h in the dark at 4°C (Parsons

et al., 1984). Fluorescence of the extracted pigments was measured on board with a 10-005R Turner Designs fluorometer [...].".

P5, L23-24: This is slightly confusing. Stored in liquid nitrogen (-196◦C) or kept frozen at -80◦C? Response: Duplicate 4 ml subsamples were fixed with 20 $\mu$l of 25% glutaraldehyde Grade I (0.1% final concentration; Sigma-Aldrich G5882), then subjected to quick-freeze in liquid nitrogen for 24h, and finally stored at -80°C until analysis. This is now added in the manuscript.

P5, L27: Did you consider sampling multiple depths in the melt-ponds? Would you have expected homogeneity or a vertical gradient? Please quickly discuss this in the text. Response: Strong saline stratification can be found in Antarctic (terrestrial) ponds but seems to be specific to deep ponds (0.5-1.5 m) (Wait et al., 2006). Jung et al. (2015) also report highly stratified open melt ponds (i.e. melt ponds that have melted all the way to the sea surface) in Arctic FYI. However, closed FYI melt pond modelling suggests that convective- and wind-driven- mixing generate well-mixed melt ponds and stratification is not a significant factor in melt pond circulation (Skyllingstad and Paulson, 2007). Some temporary stratification might be expected in the melt ponds in the absence of wind but this is rapidly (a few hours) overturned by solar heating-driven convection. The following information was added in the paragraph: "Stratified open melt ponds (i.e. melt ponds that have melted all the way to the sea surface) were reported in Arctic FYI (Jung et al., 2015). However, closed FYI melt pond, such as those sampled during this study, are not prone to vertical stratification due to convective- and wind-driven- mixing (Skyllingstad and Paulson, 2007). Given their shallow depths (less than 0.3 m), melt pond stratification was most probably inexistent or minimal during our study.".

P5, L30: "to fill the glass serum bottles" remove the "the". Response: "the" was removed

P6, L11: Consider cutting in two sentences. "...into 5 ml FalconTM tube. DMSPd was

quantified ...". Response: Sentence was cut in two as suggested.

P6, L13: Please provide whenever possible an estimate of the error associated with every measurement. This is clearly missing for the measurement of DMS(P) concentrations. Response: Measurement error estimates were added to the text.

P6, L16: Dacey and Blough (1987) is perhaps a better reference here than Levasseur et al. (2006). Response: The reference was changed in the text as suggested.

P6, L26: "freshwater", do you mean milliQ water? Please specify. Response: Yes, milliQ water is now specified in the text.

P6, L30: Consider using "duplicate" instead of "duplicated". Response: "Duplicated" was changed to "duplicate".

P7, L10: This is I think the first time a Table is mentioned in the text. It should then be Table 1. I suggest to add a reference to Table 1 earlier in the text, in section 2.1. Response: We kept the original Table 1 but mention it earlier, in section 2.1. (Table 1 " Table 1: Physical characteristics of the sea ice surrounding the melt ponds. [...].) The first mention of Table 2 is lower in the test, in the paragraph preceding section 2.2.

P8, L5: Is any fractionation expected during storage? Response: Isotopes fractionation may be caused by differences in rates of reaction or diffusion, or by differences in equilibrium constants. Fractionation during prolonged (several months) storage has been noted before in nitrogen cycle studies, even for frozen samples (Thayer, 1970; Granger et al., 2006). According to kinetics theory, kinetic energy (K.E.) is the same for all gases at a given temperature, which can result in greater velocities of lighter isotopes compared with their heavier counterparts (Sharp, 2007). Detailed calculation is provided in the Supplements to the response to referee #2 in the interactive comments section of Biogeosciences Discussion. We find that average velocity of DMS (m/z 62) is 0.8% greater than the average velocity of DMS (m/z 63) molecules in the same system. Following the same calculation steps, average velocity of DMS (m/z 62) is 4.7%

greater than the average velocity of DMS (m/z 68) molecules in the same system. Finally, average velocity of DMS (m/z 63) is 3.8% greater than the average velocity of DMS (m/z 68) molecules in the same system. Accordingly, a negligible maximum fractionation of 5% is expected during storage. Preserved samples and standards were compared against standard curves and fractionation during storage was not observed.

P8, L6-10: Please provide the overall precision of the methods. Response: The precision of the method was provided.

P9, L5: Please add this 5 m information in the section 2.1 of the materials and methods part. Response: The distance information was added to the method section.

P9, L6-8: Following my previous comments, this section needs to be rewritten. Also refrozen snow at the surface means superimposed ice, an ice texture known to be impermeable. This should be mention somewhere in the text. Response: Both reviewers pointed out the shortcomings of the section dealing with ice physics. To address this, full ice depth temperature and salinity profiles are now presented. Corresponding brine volume fraction were calculated, and presented alongside with salinity and temperature profiles in an additional figure (see new figure 3). Calculated brine volume fractions are now used to discuss sea ice permeability. The method section was also amended accordingly to reflect this additional dataset. We also mention that the refrozen snow observed at station Ice3 was impermeable and may be indicative of refreezing metamorphosis of snow.

P9, L29: Please replace (see discussion) by "This will be discussed in section...". Response: The text was modified ton "This will be discussed in sect. 4.2.2.".

P11, L11: The use of "significantly" implies a statistical test which is not provided. Response: Statistical test is now provided in the text as follows: "During both Ice1-MP1 and Ice4-MP1 incubation experiments, the Light versus Dark Treatment had no effect on the net changes in DMSPd concentrations between the L-DMSP/O and D-DMSP/O Treatments (Wilcoxon Signed-rank test; n=8, df=3, $\alpha$=0.05), but significantly

impacted the rates of net accumulation of DMS (Wilcoxon Signed-rank test; n=12, df=5, $\alpha$=0.05). The accumulation of DMS over 24h in the L-DMSP/O Treatments were consistently and significantly lower than in the corresponding D-DMSP/O Treatments (Wilcoxon Signed-rank test; n=8, df=3 , $\alpha$=0.05).".

P11, L17-30: You could make the paragraph a little bit lighter to read and easier to follow by removing some unnecessary instances of (m/z 68) and (m/z 62). Response: As suggested, several mentions of "m/z" have been removed in the paragraph.

P12, L1-2: See my previous comment. Please read the study of Leck and Persson (1996), cited in Levasseur (2013). There is also some interesting work in glacial melt water ponds that you could consult and perhaps cite somewhere in the manuscript (De Mora et al., 1996), especially regarding to DMSO as a source of DMS. Response: Studies by Leck and Persson (1996), Sharma et al. (1999), Nomura et al. (2012), and Asher et al. (2011) are now cited regarding DMS concentrations in sea ice melt ponds. De Mora et al. (1996) is cited in the introduction.

P12, L5: As stated before, I think this sentence is misleading and should be remove giving the fact that you provide further in the text a very plausible explanation for the difference. This explanation is moreover relatively logic for someone with a basic knowledge of sea ice biogeochemistry. Response: The sentence stating that "Our current limited understanding of the mechanisms responsible for the cycling of DMS in melt ponds prevents the identification of the underlying causes of these differences" was removed from the manuscript.

P12, L14: What do you mean by "closed melt pond"? It seems that the melt-pond is exchanging material with seawater and the atmosphere. Please clarify. Response: Closed melt pond terminology refers to "closed bottom" light blue coloured melt ponds that form on relatively thick ice cover. This is the only type of melt ponds discussed in our study. Open melt ponds on the contrary are dark blue ponds directly connected to the underlying seawater with a visible hole in the relatively thin ice cover. This

terminology was borrowed from Lee et al., 2012.

P12, L17 and 23: "Sea spray". Response: The typo was corrected.

P12, L27 – P13, L14: This whole section needs some rewriting. Full-depth gravity drainage should not be confused with flushing of surface melt-water. You should read a little bit more carefully the study of Jardon et al. (2013), but also Carnat et al. (2013) which describes the seasonal evolution of sea ice salinity (and brine salinity) in FYI in the Canadian Arctic (Amundsen Gulf, Beaufort Sea)[...]. Response: Comment on paragraph P12, L27 – P13, L14 was extensive and called for a re-writing of the full paragraph so we respond to each point separately in the following section: Proper terminology (brine flushing versus brine drainage) is used in the corrected version. As discussed in the response to referee #1 (P13, L10), calculations provided by Jardon et al. (2013) deal with the permeability threshold of sea ice with salinity greater than 5. With a bulk sea ice salinity of 2.79 (averaged for the three stations), and a maximum value of 5.00 at station ice4 (1.2-1.3 m section), our melt ponds fall outside of this range. The statement regarding full ice depth desalination referencing Jardon et al. (2013) was removed from the manuscript.

[...] Also, you definitely need to include brine volume fraction, Rayleigh number, and brine salinity here in the discussion. Unfortunately you only measured surface ice salinity and temperature, while full-depth profiles are generally necessary for this type of discussion. For instance, you could have 10 cm of sea ice with a low salinity due to percolating melt water with more saline layers underneath. Full-depth gravity drainage/convection requires both a connected brine network (sea ice permeable to fluid transport), and hence usually brine volumes above 5%, and an unstable brine density (brine salinity) profile [...]. Response: We took full ice depth temperature and salinity profiles during the study. We chose to only use the upper 0.1 m measurements in sea ice in the submitted version of the manuscript to illustrate the physical conditions of the ice closest to the melt ponds. Given that we discuss ice physics and permeability, we agree that it is necessary to include the full ice profiles (salinity, temperature,

and brine volume) in the revised version. These results are now presented in a new figure 3 and discussed in the revised manuscript. Averaged values for bulk sea ice salinity over the full thickness of the ice were 1.73, 2.83 and 3.75 at stations Ice1, Ice3 and Ice4, respectively. Locally, maximum bulk salinity never exceeded 5.00 (Ice4, 1.2-1.3 m section). In situ temperatures, averaged over the full thickness of the ice, were -0.54 C, -0.52 C and -0.98 C at stations Ice1, Ice3 and Ice4, respectively, and reached a minimum value of -1.39 C (Ice4, 0.8-0.9 m section). Brine volume fraction constantly exceeded 10% in the ice profiles, except in the upper 0.1 m section of the Ice3 station where we likely observed the effects of refreezing metamorphosis of snow and / or sea ice recrystallization (as mentioned in the response to P5, L1-3).

[...] The combination of these two criteria can be expressed via a Rayleigh number. When sea ice warms up and reach the permeability threshold (expressed by the brine volume fraction, not the temperature), instability of the brine network (brine salinity being a direct function of sea ice temperature (Cox and Weeks (1983)), colder surface ice has saltier and denser brine than warmer bottom ice) can result in full-depth convection, brine being replaced by upward moving seawater. This usually occurs in mid-late spring (see the study of Carnat et al. (2013)) and results in some desalination of the ice cover (the upward moving seawater being less saline than the brine it is replacing). Following further warming in summer, surface melt water (melting snow or melting surface sea ice) percolates within the brine net work leading to the process called flushing. This further decreases the bulk ice salinity down to values way under 2 as observed in your study. Warming will also dilute brine with pure ice melt water. I think that at the time of your sampling (based on the limited salinity and temperature data available), both full-depth gravity drainage and some flushing have already occurred. Hence, brine cannot indeed be responsible for the salinity observed in the melt-ponds. [...]. Response: Because of the apparent advanced desalination of sea ice in the ice cores presented, we did not include the Rayleigh (Ra) number results. We present the detailed calculation in the supplement to the response to referee #2 in the interactive comments section of Biogeosciences Discussion. Because of the high in situ ice temperatures and the low brine salinities, two terms used in Ra computation, we found negative values of Ra. Given that errors in Ra are largest for warm and permeable sea ice (Vancoppenolle et al., 2013), we decided to exclude these calculations in the reviewed version of the paper. With winter gravity drainage, flushing is the dominant desalination process for fully formed sea ice. Flushing is the three dimensional (i.e. both vertical and laterally in all directions) washing out of salty brine from the structure of porous sea ice and its replacement with a mix of seawater and melt water (Hunke et al., 2011). With our averaged bulk salinity < 4.00 throughout the ice, we agree that sea ice had most likely undergone full-depth salinity drainage and brine flushing before our sampling. New paragraph: "Ice brine intrusion is also unlikely to have contributed significantly to melt pond salinization since the averaged bulk ice salinity was low (under 5), especially in the top 0.2 m where it did not exceeded 2. It is also known that most of the hyper-saline brine characterizing consolidated cold FYI in winter are lost in spring through full depth brine convection well before melt ponds start to form (Jardon et al., 2013). Residual salts are finally lost through meltwater flushing during the summer season (Weeks and Ackley, 1986, Eicken et al., 2002; Vancoppenolle et al., 2007). At the time of our sampling, low bulk salinity values, combined with calculated brine volume fraction constantly exceeding 10% in the entire sea ice profiles (except in the upper 0.1 m section of the Ice3 station) suggest that full depth flushing had already occurred. We thus exclude sea ice brine enrichment of melt ponds as their main salinization mechanism.". [...] Now you still have to explain how to get seawater in contact with the melt-pond water through the porous brine network. Full-depth gravity drainage as suggested P13L10 makes no sense to me as the brine salinity do no support instability anymore. You also have to be a little bit careful with the use of the freeboard, especially citing Hudier et al. (1995). What Hudier et al. (1995) refers to is the loading of the sea ice surface with a significant amount of snow, depressing the surface sea ice level below the seawater level, leading to flooding of the ice surface, followed by gravity drainage. This is not really what you observed here. I agree that the decrease in sea ice thickness and development of the melt pond translate into a

loss of freeboard, and that the melt-pond depth might approach the freeboard height, or even get below that height. Given the height of the freeboard and the depth of the melt-pond, seawater might infiltrate the porous ice texture via the brine network and start exchanging with the melt-pond. I am a little puzzled by the diffusion mechanism you suggest. It is probably true that at some point of the melt-pond evolution, infiltrated melt water might freeze and block the flushing of the pond by decreasing permeability in the ice layer under the melt-pond. No direct exchange with underlying seawater would then be possible. Diffusion could occur but would be a very slow process (especially through such layer), rather unlikely to explain the salinity change and biomass seeding observed in the pond. Alternatively, I wonder if the pond evolution could not alternate between phases of flushing, and phases of replenishment (pond depth being close to or below the freeboard height) with a mix of seawater and pond water. These phases would be controlled by small changes in ice temperature oscillating around the freezing temperature of the melt water. I think that the similarity in species composition between the melt-pond and under-ice seawater supports well this mechanism. Response: The paragraph was modified as follows: "Rather, we suggest that melt ponds salinization originated mostly from the intrusion of seawater through the ice. Although closed melt ponds are not visibly connected to seawater, exchanges with the underlying seawater can take place. The extent of these exchanges are dependent on the sea ice freeboard and micro-structure, i.e. the amount, size and shape of brine inclusions (Carnat et al., 2014), that controls sea ice permeability. Above a critical brine volume ranging between 5% (for columnar sea ice) and 10% (for granular sea ice), brine inclusions become interconnected. During the melting season, decrease in sea ice thickness is enhanced by the formation of the melt pond and lead to a loss of freeboard. As melt ponds become closely levelled with seawater, small changes in ice temperature oscillating around the freezing temperature may result in episodic intrusion of seawater mixed with meltwater through the porous ice. Seawater mixed with meltwater penetrating the brines channels of permeable sea ice may bring salts, nutrients and microorganisms (Jardon et al., 2013, Vancoppenolle et al., 2010), potentially reaching surface melt ponds. This

mechanism most probably explains the salinity and biochemical characteristics of Ice1 and Ice4 melt ponds. Station Ice3 represents a different case. Here, the low melt pond salinity (and absence of biological activity) may be explained by the presence of an impermeable ice layer on the top of the ice preventing both pond drainage and exchange between pond water and seawater. We also considered the following paragraph as a valuable addition to the discussion: "We acknowledge that our data set is too limited to draw firm conclusions on the processes governing the formation and salinization of FYI melt ponds. Yet, in the interest of further research, we conjecture that snow load before melt onset may be crucial in determining the fate of melt ponds not only with regards to their saline status, but also their potential to produce DMS. Brine volume, derived from bulk salinity and temperature, generally provides a valid proxy for sea ice permeability. In some case however, melting of high snowpack generates a considerable flow (up to 15cm d-1) of freshwater into the porous structure of sea ice (Polashenski et al., 2017). This can create localized ice plugs within the highly connected brine network of apparently porous sea ice and allow melt ponds to persist above sea level well after sea ice bulk sea ice brine volume reached a critical level (5-10%). Such deviation from the porosity/permeability relationship following freshwater intrusion is demonstrated in Polashenski et al. (2017). We suggest that we observed such case of melt pond persistence above sea level in station Ice3. Alternatively, lower snow load remaining at the onset of the melt season will translate into a less abundant freshwater input above sea ice. Snow load distribution is however notoriously highly variable even at the meter scale due to wind redistribution and sea ice topography variability (e.g. Polashenski et al., 2017). Low snowpack would induce limited insulation of the sea ice from atmospheric conditions, resulting in 1) a more gradual warming of sea ice during spring season, and 2) limited freshwater loading available for percolation blockage. In such case, freshwater would not seal the ice through percolation blockage (Polashenski et al., 2017). Sea ice would then remain entirely porous as soon as the 5-10% brine volume threshold is reached, facilitating melt pond salinization process. We suggest that this scenario may have been observed at stations Ice1 and Ice4.".

P14, L6-8: "over-flooding of sea ice". Replace by "flooding of the ice surface". Over-flooding is an odd term. Response: "Over-flooding" is no longer used throughout the text.

P14, L6: Flooding could be better defined. Response: Flooding was redefined in the introduction as follows: "[...] flooding, a process whereby heavy snow load pushes the ice below the water level. Flooding is common in the Antarctic and results in the formation of snow ice (Hunke et al., 2011).".

P16, L7: Again, consider other data sets available. Response: Datasets from Nomura et al. (2012) and Leck and Persson (1996) are now considered.

P16, L16: Modify "over-flooding". Response: Over-flooding was modified to flooding as suggested in P14, L6-8.

P16, L20: There are several studies providing direct (Nomura et al., (2012) and indirect (Carnat et al., (2014)) evidences of DMS flux from FYI surface toward the atmosphere. Response: These references were added to the manuscript. The text now reads: "To this day, most climatologies assume the absence of DMS fluxes above ice-covered waters (e.g. Lana et al., 2011) even though several studies provide direct (Zemmelink et al., 2008; Nomura et al., 2012, MYI) and indirect (Carnat et al.,2014, FYI) evidence of DMS venting from snow-covered Antarctic sea ice.".

P16, L24: These numbers should be put in perspective. How do they compare to the sea ice, ocean reservoirs? Response: As explained previously (response to comment P2, L11-15), we deleted the estimate of the size of DMS reservoir in Arctic FYI melt ponds from the manuscript.

P16, L26: Is the average depth calculated from your data set or from literature observations? Your data set is relatively small. Response: This is the average depth for the 9 melt ponds measured during this study. We agree that a greater spatial coverage of melt ponds is needed to come up with a more robust estimate and we therefore

deleted the section of the discussion (and abstract) where we tentatively estimated the contribution of melt ponds to the overall size of the DMS reservoir in Arctic.

P16, L29: Wind velocity but also a better understanding of gas exchange between small fetch melt ponds and the atmosphere. Response: The necessity of a better understanding of gas exchange between small fetch melt ponds and the atmosphere was added to the manuscript.

References: Check the alphabetic order, Giamarelou et al. should be after Garrison. Response: The alphabetical order of references was checked and corrected.

Figures and tables

Table 2: check the significant digits in the temperature values. Only physical characteristics are presented here, remove the chemical and biological characteristics from the caption. Response: Temperature values are now expressed with only one significant digit in Table 2 and the title of the table was corrected as suggested.

Table 7: Please be consistent with the significant digits. Response: Done.

Figure 1: Please add a scale on figure 1b. As requested above, it would be nice to indicate the melt-pond fractions on each picture and an explanation of the calculation in the caption. Response: Scales were added on figures 1b and 1c along an estimate of pond fraction.

Figure 3: Odd lettering of the figures. Response: (*Now figure 4) The numbering was modified so that "c" and "b" are now interchanged.

[revised manuscript text omitted]

Please also note the supplement to this comment:
https://www.biogeosciences-discuss.net/bg-2017-432/bg-2017-432-AC2-supplement.pdf

[Figure]

**Supplement:**

**Supplementary material to the response to referee #2 in the interactive comments on "Dimethylsulfide dynamics in first-year sea ice melt ponds in the Canadian Arctic Archipelago"**

Margaux Gourdal et al.

margaux.gourdal@takuvik.ulaval.ca

This material is to be viewed in complement to the response to the comments of the second referee on the manuscript bg-2017-432 (https://doi.org/10.5194/bg-2017-432) under revision in Biogeosciences Discuss.

Here, we detail the calculations used to reply to the specific comments P8, L5 and P12, L27 – P13, L14 .

**Specific comment P8, L5**: Is any fractionation expected during storage?

**Response:** Isotopes fractionation may be caused by differences in rates of reaction or diffusion, or by differences in equilibrium constants. Fractionation during prolonged (several months) storage has been noted before in nitrogen cycle studies, even for frozen samples (Thayer, 1970; Granger et al., 2006). According to kinetics theory, kinetic energy (K.E.) is the same for all gases at a given temperature, which can result in greater velocities of lighter isotopes compared with their heavier counterparts (Sharp, 2007). Applying K.E. equations to the isotopes of interests (e.g. DMS m/z 62 and 63), this would result in the following calculation:

$$K.E. (DMS\ m/z\ 62) = K.E. (DMS\ m/z\ 63) \tag{1.1}$$

and
$$K.E. = \tfrac{1}{2}mv^2 \tag{1.2}$$

where m is mass and v is velocity. Substituting the masses of the DMS isotopes equation (1.1) becomes

$$\tfrac{1}{2}(62)(v62)^2 = \tfrac{1}{2}(63)(v63)^2 \tag{1.3}$$

$$v_{62} = \sqrt{(63/62)}v63 = 1.008 v_{63} \tag{1.4}$$

Average velocity of DMS (m/z 62) is 0.8% greater than the average velocity of DMS (m/z 63) molecules in the same system. Following the same calculation steps, average velocity of DMS (m/z 62) is 4.7% greater than the average velocity of DMS (m/z 68) molecules in the same system. Finally, average velocity of DMS (m/z 63) is 3.8% greater than the average velocity of DMS (m/z 68) molecules in the same system.

According to the above calculations, a negligible maximum fractionation of 5% is expected during storage. Preserved samples and standards were compared against standard curves and fractionation during storage was not observed.

**Specific comment P12, L27 – P13, L14:** [...] you definitely need to include brine volume fraction, Rayleigh number, and brine salinity here in the discussion [...].

**Response:** Because of the apparent advanced desalination of sea ice in the ice cores presented, we did not include the Rayleigh (Ra) number results. We still present the calculation in the following paragraph intended only as a response to the specific comment P12, L27 – P13, L14 in order to clarify our process.

Sea ice can be mathematically described as a mushy-layer medium (e.g. Hunke et al., 2011) a multicomponent multiphase medium where interstitial liquid can flow within the porous medium (e.g. Worster and Kerr, 1994 ). Borrowed from the

metallurgy industry, mushy layer theory has been increasingly adopted in sea ice studies because it advantageously provides one set of equations for various sea ice micro-structure configurations. Within the mushy-layer framework, calculation of Ra allows for the description of the onset and strength of gravity drainage (winter). Vertical density gradients, caused by the presence of highly salted brines on the top layers of sea ice, cause instability within the mushy-layer. Ra can thus be used as a proxy for the onset and the intensity of gravity drainage (Carnat et al., 2013). Multiple parameterizations of Ra can be found in the literature (Notz et al., 2009, Carnat et al., 2013 ; Vancoppenolle et al., 2013). They all yield different results leading to different interpretation of ice core data (Vancoppenolle et al., 2013). Field based observations suggest that full depth gravity drainage can be initiated when Ra value reaches 3 (Carnat et al., 2013). A Ra threshold value greater than 10 as suggested in Notz and Worster (2008) is most likely too high for brine convection in situ. Nevertheless, Ra addresses sea ice undergoing gravity drainage, a winter process that influences brine-rich sea ice. Ra has already been used in studies including warming sea ice (e.g. Carnat et al., 2013), but reported values of bulk sea ice salinity and temperature indicate a less advanced stage of sea ice melt in Carnat et al. (2013) than here. During our study, we most likely passed the point of full-depth desalination of warming sea ice given the low salinities observed throughout the ice profiles (Fig. 3). For such highly permeable sea ice, brine loss during sampling may lead to an underestimation of bulk salinity up to 20 g kg$^{-1}$ (Notz et al., 2005) and thus increase the uncertainty of Ra number calculation (VanCoppenolle et al., 2013).

Ra was calculated for sea ice (T > -2°C) following the formula presented in Carnat et al. (2013):

$$Ra = \frac{\Delta z g \beta (S_b(z) - S_{OC}) \Pi (V_b/V_{min})}{\kappa \eta}.$$

As reported in Carnat et al. (2013) : $g = 9.81$ m$^{-2}$ is the acceleration due to gravity; $\beta = 0.78$ kg m$^{-3}$ ppt$^{-1}$ is the haline expansion coefficient of sea water at 0°C (Fofonoff, 1985); $S_{b\,(z)}$ is the salinity of brine at depth z (depth increasing from zero in the bottom ice to the top of sea ice) calculated using Lepparanta and Manninen (1988), z is given as the distance from a specific level in the ice to the ice-ocean interface ; $S_{OC}$ is the salinity of sea water; $\Pi(V_b/V_{min})$ is the effective sea ice permeability (m$^2$) calculated using the formula below (Notz et al., 2009 parametrization of $\Pi$ was also used for comparison) ; $\kappa = 1.3$ x $10^{-7}$ m s$^{-2}$ is thermal diffusivity of brine (Sharqawy et al., 2010) ; and $\eta = 2.55$ x $10^{-3}$ kg m s$^{-1}$ is the dynamic viscosity of brine.

The formula used to calculate the effective sea ice permeability $\Pi$ was found in the technical discussion paper by Vancoppenolle et al., 2013 (www.the-cryosphere-discuss.net/73209/2013/), after Eicken et al., 2004:

$$\Pi = \begin{cases} 4.708 \times 10^{-14} \cdot \exp(76.90e), & e \leq 0.096 \\ 3.738 \times 10^{-11} \cdot \exp(7.265e), & e > 0.096. \end{cases}$$

*e* expresses the brine volume fraction and is calculated as follows:

$$e \approx S/S_{br}$$

Where $S$ is bulk ice salinity and $S_b$ is brine salinity calculated using Lepparanta and Manninen (1988). In our study, *e* values ranged between 0.113 and 0.478. Accordingly, the second equation for sea ice permeability $\Pi$ was used. Because of the high in situ ice temperatures and the low brine salinities, two terms used in Ra computation, we found negative values of Ra. Given that errors in Ra are largest for warm and permeable sea ice (Vancoppenolle et al., 2013), we decided to exclude these calculations in the reviewed version of the paper.

With winter gravity drainage, flushing is the dominant desalination process for fully formed sea ice. Flushing is the three dimensional (i.e. both vertical and laterally in all directions) washing out of salty brine from the structure of porous sea ice and its replacement with a mix of seawater and melt water (Hunke et al., 2011). With our averaged bulk salinity $< 4.00$ throughout the ice, we agree that sea ice had most likely undergone full-depth salinity drainage and brine flushing before our sampling.